# Reinventing Policy Iteration under Time Inconsistency

**Nixie S. Lesmana** *nixiesap001@e.ntu.edu.sg*
*School of Physical and Mathematical Sciences*
*Nanyang Technological University*

**Huangyuan Su** *huangyus@andrew.cmu.edu*
*School of Computer Science*
*Carnegie Mellon University*

**Chi Seng Pun** *cspun@ntu.edu.sg*
*School of Physical and Mathematical Sciences*
*Nanyang Technological University*

**Reviewed on OpenReview:** *https://openreview.net/forum?id=bN2vWLThOP*

## Abstract

Policy iteration (PI) is a fundamental policy search algorithm in standard reinforcement learning (RL) setting, which can be shown to converge to an optimal policy by policy improvement theorems. However, the standard PI relies on Bellman's Principle of Optimality, which might be violated by some specifications of objectives (also known as time-inconsistent (TIC) objectives), such as non-exponentially discounted reward functions. The use of standard PI under TIC objectives has thus been marked with questions regarding the convergence of its policy improvement scheme and the optimality of its termination policy, often leading to its avoidance. In this paper, we consider an infinite-horizon TIC RL setting and formally present an alternative type of optimality drawn from game theory, i.e., subgame perfect equilibrium (SPE), that attempts to resolve the aforementioned questions. We first analyze standard PI under the SPE type of optimality, revealing its merits and insufficiencies. Drawing on these observations, we propose backward Q-learning (bwdQ), a new algorithm in the approximate PI family that targets SPE policy under non-exponentially discounted reward functions. Finally, with two TIC gridworld environments, we demonstrate the implications of our theoretical findings on the behavior of bwdQ and other approximate PI variants.

## 1 Introduction

Policy iteration (PI) has enjoyed a long history of success in standard reinforcement learning (RL), which can be attributed to standard PI that combines a dynamic programming (DP)-based policy evaluation[1] and a greedy policy improvement; see Bellman (1957); Howard (1960). Standard PI has been the basis of many classical RL algorithms, such as value iteration and the popular Q-learning (Watkins & Dayan (1992)), and it still inspires the design of modern RL algorithms. Despite its prominence in standard RL setting, standard PI has been deemed incompatible for time-inconsistent (TIC) objectives due to *non-monotonicity* and the implied *violation of Bellman's principle of optimality (BPO)*.

Time inconsistency (also abbreviated as TIC) is prevalent in dynamic choice problems and captures well human's tendency to deviate from their current plan at a future time; such deviation arises as a plan of future course of actions that is *optimal* for a human agent *today*, may *not* be *optimal* for the same agent in the *future*. In the context of RL, TIC often arises as an effort to more closely model human preferences, resulting in TIC objectives upon which an RL agent is built on, and has been investigated through several major channels such as hyperbolic discounting and risk-sensitive RL.

---

[1]For a formal introduction, readers may refer to Eq (8).

The idea of questioning the validity of standard PI under TIC was pioneered in risk-sensitive RL by Sobel (1982). In this seminal work, a counterexample to the monotonicity[2] property (also referred to as consistent choice, temporal persistence, or stationarity across the literature) was posted and attention was raised in how this property is commonly exploited to prove the convergence of standard policy improvement scheme to an optimal policy. Two puzzles are then left for answers:

**How optimal is the termination policy (the policy obtained at the end of an algorithmic search) of standard PI?**

**and**

**Is it possible to guarantee the update monotonicity (a desirable algorithmic property that will lead to convergence) of standard PI?**

In this paper, we focus on infinite-horizon TIC RL problems and formally present the subgame perfect equilibrium (SPE) notion of optimality that corresponds to how sophisticated, rational agent acts in the face of TIC i.e., planning consistently in terms of solving optimizations that take into account the future deviations. We will then revisit the two questions above to highlight standard PI's merits and insufficiencies in achieving the SPE notion of optimality.

The contribution of this paper can be summarized as follows:

- In terms of optimality, we establish that the termination policy of standard PI under TIC achieves SPE.

- We study the failure of policy improvement theorem (Sutton & Barto (2018)) and highlight some insufficiencies of standard PI update and the existing analysis tools, in the context of SPE policy search.

- TIC-adjusted DP formula is established to compute nonexponentially-discounted Q-function, addressing the insufficiency of standard DP formula.

- Based on the aforementioned analyses, we devise a new PI paradigm for non-exponentially discounted reward functions: backward Q-learning (bwdQ).

- We design toy Gridworld examples to demonstrate the implications of our findings on the behaviour of bwdQ and other approximate PI variants under TIC.

- The analyses (in Section 5.1 and 4.3) relevant to the advantage of *backward conditioning* in bwdQ is of independent interest: the characterization of its termination policy as SPE and its efficiency-related desirability as an SPE policy learner extend beyond general-discounting objectives.

Note that some lengthy proofs/justifications of our results are deferred to Appendix.

## 2 Related Works

**Non-monotonicity in risk-sensitive RL and solutions.** In risk-related context, several follow-up works since Sobel (1982) address the non-monotonicity issue following the line of reasoning that the search for a globally-optimal policy in non-monotonic problems are computationally expensive (as one can only enumerate over the whole policy space that is almost impossible in practice) and hence, new solutions are desired. For instance, Mannor & Tsitsiklis (2011) formally compares between several policy classes to reduce the search problem for globally-optimal policy (to a specific policy class) and proposes several practical approximation algorithms. One important finding in their work is that randomization can improve control performance;

---

[2]A formal definition to monotonicity is provided in Eq (15). Intuitively, if a decision-making problem is monotonic, then delaying the use of any two decision policies will preserve their ordering (in terms of the policies' values from any states). When this property holds, we can typically break our problem into subproblems as in BPO and attain computational efficiency.

this inspires Di Castro et al. (2012); Tamar & Mannor (2013); Prashanth & Ghavamzadeh (2013) to propose gradient-based algorithms accustomed to mean-variance criteria, which highlighted parameterized stochastic policy as a manner to deal with non-monotonicity. The latter works are relevant to our case as they also use TIC adjustment terms to obtain temporal difference (TD)-based policy evaluation (PE) that resembles the one used in extended DP theory Björk et al. (2014). To distinguish our approach, we note our focus on using SPE policy itself to deal with non-monotonicity (by *modifying our optimality type*) as opposed to randomization or parameterization.

**Non-monotonicity in hyperbolic-discounting RL and solutions.** In hyperbolic-discounting context, non-monotonicity have also appeared, independent of Sobel (1982)'s work; see Kurth-Nelson & Redish (2010) for instance. In this work, several proposals towards computationally practical models are reviewed, with varying action selection strategies drawn largely from behavioral or neuroscience point of view. A recent follow-up work by Fedus et al. (2019) extends their distributed micro-agents model (i.e. $\mu Agents$) to handle larger scale problems, utilizing deep neural network to model the different Q-values from a shared representation. Though such modifications in action selection may have implicitly addressed the non-monotonicity underlying PIT failure, to the best of our knowledge, an explicit connection between the two (as in Sobel (1982)) has never been made.

**Time-consistent Planning and Control.** The idea of locally optimal, time-consistent planning under TIC was pioneered by Strotz (1955); Pollak (1968). This type of planning corresponds to a sophisticated, rational agent's behavior who, when faced with TIC, compromises with their future selves by taking future disobedience as a constraint in their decision-making. The solution concept is developed as a game-theoretic framework that builds on backward inductive SPE search in games, thus coining the term SPE plan or policy. This then leads to an *intra-personal equilibria* formalism by Björk & Murgoci (2014) which unifies several task-specific TIC sources through extended DP theory and has attracted a wide array of literature in TIC stochastic control. The rise of SPE policy as a major contending solution to the globally optimal (precommitment) policy can then be attributed to two reasons: (i) as a controller, precommitment policy may lead to some undesirable outcomes since it may lose its optimality as time evolves (for instance, due to an unpredictable change in environment dynamics), (ii) computationally, there is lack of a pivotal tool to identify a globally-optimal policy that generalizes naturally to different TIC tasks (for instance, due to its disconnection to standard DP that requires BPO).

**SPE Policy in TIC-RL.** Some works in the general-discounting space have investigated TIC-RL from a purely behavioral lens, focusing particularly on the property of target policy rather than a computational aspect. For instance, Lattimore & Hutter (2014) proposes rational agents that act according to history-dependent SPE policies. In this work, the authors cover some theoretical aspects of policies such as characterization of different policy types, existence results connecting discounting and policy types, and comparative study in some example scenarios. In another work, Evans et al. (2016) proposes sophisticated agents that act according to Markovian SPE policies and are modelled with delay-augmented Q-learning algorithms. Though relevant, these algorithms are proposed in the context of generative models that aids human-like preference inferences; thus, algorithmic properties are not covered. A recent work by Lesmana & Pun (2021) considers the search of Markovian SPE policy under finite-horizon task-invariant TIC objectives. Drawing inspiration from the extended DP theory, the authors propose Backward Policy Iteration (BPI), which has lex-monotonicity guarantee in place of the standard policy improvement theorem. This work is the closest to ours where our *backward conditioning* can be viewed as an infinite-horizon extension to BPI. We distinguish our contribution by noting our main focus on *analyzing standard PI*, that motivates our *infinite-horizon, Markovian* SPE policy formalism and the corresponding drop of time-dependency, *shifting definition of players from times to states*. Relative to finite-horizon case, such formalism introduces technical challenges in both aspects of policy evaluation and improvement, which we will remark on the respective sections of this paper.

## 3 Problem Formulation and the SPE Concept

In this section, we introduce the class of TIC RL problems of our interest and formally present the solution concept of SPE policy. We then cast the general-discounting objective as a TIC RL problem and construct a few examples in this context that we will quote frequently throughout the paper.

### 3.1 TIC RL Problem Formulation

We consider the policy search in an infinite-horizon TIC-MDP, which consists of the standard MDP tuple $(\mathcal{S}, \mathcal{A}, \mathbb{P}, \mathcal{R})$ and a specific TIC source. The state space $\mathcal{S}$ and action spaces $\mathcal{A}_s \subseteq \mathcal{A}, \forall s \in \mathcal{S}$, are assumed to be discrete and finite with stationary probabilities $p_s^a(\cdot) := \mathbb{P}[R_{t+1} = \cdot, S_{t+1} = \cdot \mid S_t = s, A_t = a]$ governing the transitions from a current state $S_t = s$ to the next state $S_{t+1}$ and reward $R_{t+1}$ for $s \in \mathcal{S}$, given a particular action $A_t = a$. To define a stopping criterion, it is convenient to augment a so-called *absorbing* state, denoted by $\bar{s}_{\text{void}}$, which incurs no reward. Then, we define a *stopping* action $\bar{a}$ as an action that drives a transition to $\bar{s}_{\text{void}}$ from any states $s \in \mathcal{S}$ and *boundary* states $\bar{s} \in \bar{\mathcal{S}}$ as rewarding states with specific action space $\mathcal{A}_{\bar{s}}$ or $\bar{\mathcal{A}} := \{\bar{a}\}$, i.e. once the boundary state is reached, we conclude with reward as there is only action $\bar{a}$ that will transit us from $\bar{s}$ to $\bar{s}_{\text{void}}$ and then make us stay at $\bar{s}_{\text{void}}$ forever. This setup is to complete the mathematical framework of the environment for analyses, where the problem of interest has certain stopping criteria, e.g., after receiving a target reward.

Let us next denote by $\Pi^{MD}$ the set of all Markovian, deterministic policies $\boldsymbol{\pi} := \{\pi(s) : s \in \mathcal{S}\}$ with $\pi : \mathcal{S} \to \mathcal{A}_s$. To aid presentation in subsequent sections, we define $a \cdot \boldsymbol{\pi}$ as a policy that prescribes the use of action $a \in \mathcal{A}$ for a current one-step decision and policy $\boldsymbol{\pi}$ for all remaining decisions. Similarly, we denote by $\boldsymbol{\delta}^\tau \cdot \boldsymbol{\pi}$ a policy that fixes the first $\tau > 0$ steps decisions to $\boldsymbol{\delta}^\tau := \{\delta(S_w) : t \le w < t + \tau\}$, with a map $\delta : \mathcal{S} \to \mathcal{A}_s$ and a current time $t$, and follows $\boldsymbol{\pi}$ afterwards.

**TIC reward structures and criterion** We first note that by our assumption on stationary transitions, we are limiting our TIC scope to those that arise from reward structures and criterion, described as follows. Let us consider a general criterion $V^{\boldsymbol{\pi}}(s)$ (with form not restricted at this point) for any $\boldsymbol{\pi} \in \Pi^{MD}$ and $s \in \mathcal{S}$. Given an initial state $s_0 \in \mathcal{S}$, a standard notion of optimality aims to solve the *global* problem $\mathcal{P}_{0,s_0} \doteq \max_{\boldsymbol{\pi}} V^{\boldsymbol{\pi}}(s_0)$ and obtain the corresponding globally-optimal (precommitment) policy denoted by $\boldsymbol{\pi}^{*0}$. Next, let us define for each delay $\tau > 0$, the *local* problem $\mathcal{P}_{\tau,s_\tau} \doteq \max_{\boldsymbol{\pi}} V^{\boldsymbol{\pi}}(s_\tau)$, where $s_\tau$ represents any realization of $S_\tau$ following the sequence of policies $\{\pi^{*0}(S_t) : 0 \le t < \tau \mid S_0 = s_0\}$, and denote by $\pi^{*\tau}(s_\tau)$ its solution. **Bellman's Principle of Optimality (BPO)** then states that

$$\forall \tau, s_\tau, \ \pi^{*\tau}(s_\tau) = \pi^{*0}(s_\tau) \tag{1}$$

By the BPO definition above, the standard RL criteria belong to the time-consistent (TC) class that does not violate (1). While in this paper, we consider criteria $V^{\boldsymbol{\pi}}(s)$ that violate (1); these include general-discounting, risk-related, and more (cf. Björk & Murgoci (2014)). Given any criteria $V^{\boldsymbol{\pi}}(s)$, one can verify whether it belongs to the TIC class through a counterexample (which will be illustrated with Example 3.7 below).

It is noteworthy that our way in defining TIC criterion is unlike most MDPs that specify the criterion $V^{\boldsymbol{\pi}}(s)$ up to the expectation of cumulative rewards. We aim to maintain generalities up to the formalism of SPE optimality (in Section 3.2) that is valid for different forms of TIC reward structures and criterion. For instance, while general-discounting objectives still admit an expectation form, risk-sensitive objectives involve non-linearity in expectations. That said, to fully define a TIC-MDP, we need the exact specifications of reward structures $\mathcal{R}$ and the corresponding TIC sources. We will further discuss on this topic in Section 3.3 with general-discounting specifications.

### 3.2 SPE Notion of Optimality

In the previous subsection, we have established our focus on the objectives that violate BPO in (1). Once BPO is violated, $\boldsymbol{\pi}^{*0} \ne \boldsymbol{\pi}^{*\tau}$ for some $\tau, s_\tau$ and we will have a collection of *competing optimization problems* $\{\mathcal{P}_{\tau,s_\tau} : \forall \tau \in [0, \infty), s_\tau \in \mathcal{S}\}$ to solve. We then have two options. Firstly, we can focus on only a single agent corresponding to $(0, s_0)$, denoted by Agent-$s_0$, and solve for a globally optimal or so-called

precommitment policy. However, as we have noted in Section 2, this is expensive to obtain in general and more importantly, it has been known to suffer from the two puzzles in Section 1. Secondly, we can consider $\{\mathcal{P}_{\tau,s_\tau} : \forall \tau \in [0,\infty), s_\tau \in \mathcal{S}\}$ as a multi-agent problem, where each Agent-$s_\tau$ is associated with the problem $\mathcal{P}_{\tau,s_\tau}$, and solve for the Nash equilibrium of this (sub-)game, i.e. SPE.

In this subsection, we will formalize such SPE notion of optimality in an *infinite-horizon* setting. Let us consider any TIC criterion $V^{\boldsymbol{\pi}}(s)$ and define the corresponding action-value or Q-function $Q^{\boldsymbol{\pi}}(s,a) := V^{a \cdot \boldsymbol{\pi}}(s)$. Our aim is then to find an (stationary) SPE policy $\hat{\boldsymbol{\pi}}$, defined as follows.

**Definition 3.1** (SPE Policy). A policy $\hat{\boldsymbol{\pi}} \in \Pi^{MD}$ is an SPE policy if it satisfies

$$Q^{\hat{\boldsymbol{\pi}}}(s, \hat{\pi}(s)) \geq Q^{\hat{\boldsymbol{\pi}}}(s,a), \forall a \in \mathcal{A}_s, \forall s \in \mathcal{S} \tag{2}$$

In other words, our game consists of Players, indexed by the *states* $s \in \mathcal{S}$, and we search for an SPE, where Player $s$ takes into account the strategies of other Players $s' \in \mathcal{S} \setminus \{s\}$ in its decision making as $s$ will be transited to an $s'$. In particular, Definition 3.1 implies that any state $s'$ does not have incentive to deviate from its strategy $\hat{\pi}(s')$ when everyone else plays $\hat{\boldsymbol{\pi}}$; note that this is how SPE takes into account future deviations, as we mentioned in Section 1.

*Remark* 3.2. It is important to note that in standard RL, the policy that realizes Definition 3.1 (or so-called non-positive advantage) is the optimal policy. However, such equivalence draws on BPO, which does not apply in TIC RL. Once BPO is violated, the optimal policy (one that solves $\mathcal{P}_{0,s_0} := \max_{\boldsymbol{\pi}} V^{\boldsymbol{\pi}}(s_0)$, that we refer to as globally-optimal/precommitment policy $\boldsymbol{\pi}^{*0}$) is no longer equivalent to the one that realizes non-positive advantage (one that solves the game induced by competing local optimizations $\mathcal{P}_{\tau,s_\tau} := \max_{\boldsymbol{\pi}} V^{\boldsymbol{\pi}}(s_\tau)$, that we refer to as SPE policy $\hat{\boldsymbol{\pi}}$). As we will see later in Section 3.3, an SPE policy (cf. Figure 1(c)) can be neither equal or equivalent in value to the globally-optimal/precommitment policy (cf. Figure 1(b)).

Throughout this paper, we will adopt several technical assumptions to address the technical challenges of such *infinite-horizon* SPE, particularly those that arise from the derivations of *TIC-adjusted DP* in Section 4.2 and *backward conditioning* update in Section 5.1.

**Assumption 3.3.** $\forall s \in \mathcal{S}, \forall \epsilon > 0$, there exists a *truncation step* $\bar{T} < \infty$ s.t. $\forall \boldsymbol{\pi} \in \Pi^{MD}, \ddot{\boldsymbol{\pi}} : \mathcal{S} \to \bar{\mathcal{A}}$,

$$\left| V^{\boldsymbol{\pi}}(s) - V^{\boldsymbol{\pi}^{\bar{T}} \cdot \ddot{\boldsymbol{\pi}}}(s) \right| \leq \epsilon. \tag{3}$$

Denote by $\bar{T}_{s,\epsilon}$ the *smallest* such $\bar{T}$.

**Assumption 3.4.** $\exists s_0 \in \mathcal{S}$ s.t. $\exists \hat{T}_{s_0} < \infty$

$$\forall s \in \mathcal{S} \setminus \{s_0\}, \exists \boldsymbol{\pi} \in \Pi^{MD}, \sum_{t=0}^{\hat{T}_{s_0}} \mathbb{P}[S_t^{\boldsymbol{\pi}} = s | S_0 = s_0] > 0.$$

Intuitively, Assumption 3.3 asserts that starting from any $s$ and following any policy $\boldsymbol{\pi}$, any rewards generated after $\bar{T}$ steps are negligible as the policy $\ddot{\boldsymbol{\pi}}$ incurs 0 rewards[3]. Then, Assumption 3.4 ensures the existence of at least one state $s_0$ from which all other states $s \in \mathcal{S} \setminus \{s_0\}$ can be reached in finite time, with positive probability. Combining these two, we fix $s_0$ and set $\bar{T}_\epsilon := \max\{\hat{T}_{s_0} + 1, \bar{T}_{s_0,\epsilon}\}$ to obtain our last assumption.

**Assumption 3.5.** Let us define $\mathcal{S}_0^{s_0, \Pi^{MD}} := \{s_0\}$ and $\forall t \in (0,\infty)$,

$$\mathcal{S}_t^{s_0, \Pi^{MD}} := \{s \in \mathcal{S} : \exists \tau \in [0,t), s_\tau \in \mathcal{S}_\tau^{s_0, \Pi^{MD}} \text{ s.t. } \exists \boldsymbol{\pi} \in \Pi^{MD}, \mathbb{P}[S_{t-\tau}^{\boldsymbol{\pi}} = s | S_0 = s_\tau] > 0\} \tag{4}$$

Then, $\forall \epsilon > 0, \forall t \in [0, \bar{T}_\epsilon], \forall s_t \in \mathcal{S}_t^{s_0, \Pi^{MD}}$, and $\forall \boldsymbol{\pi} \in \Pi^{MD}$ with $\mathbb{P}[S_t^{\boldsymbol{\pi}} = s_t | s_0] > 0$, $\exists \kappa(\epsilon) = \kappa(\epsilon, t, s_t, \boldsymbol{\pi}; s_0) > 0$ s.t. $\left| V^{\boldsymbol{\pi}}(s_t) - V^{\boldsymbol{\pi}^{\bar{T}_\epsilon - t} \cdot \ddot{\boldsymbol{\pi}}}(s_t) \right| \leq \frac{\epsilon}{\kappa(\epsilon)}$ and $\lim_{\epsilon \downarrow 0} \frac{\epsilon}{\kappa(\epsilon)} = 0$.

---

[3]We note one important implication of Assumption 3.3: bounded value functions, which proof can be found in Appendix A.2.

Intuitively, Assumption 3.5 ensures the negligibility of rewards under $\boldsymbol{\pi}$ from any intermediate states $s_t \in \mathcal{S}_t^{s_0, \Pi^{MD}}$ when there is a path connecting it to $s_0$ under the same policy $\boldsymbol{\pi}$. In Appendix A.1, we show that these assumptions are reasonable in practice through a set of sufficiency conditions (in general-discounting context, in terms of restrictions on MDP and discounting function).

### 3.3 General-discounting Criterion

As a major concern of this paper, we consider the following infinite-horizon criterion

$$V^{\boldsymbol{\pi}}(s_\tau) \doteq \mathbb{E}\left[\sum_{t=\tau}^{\infty} \varphi(t-\tau)R(S_t^{\boldsymbol{\pi}}, \pi(S_t^{\boldsymbol{\pi}}))\Big| S_\tau = s_\tau\right] \tag{5}$$

defined for any $\tau \geq 0$, with a general discounting function $\varphi : \mathbb{N} \to (0,1]$. The intermediate (possibly random) reward function $R : \mathcal{S} \times \mathcal{A} \to \mathbb{R}^+$ follows the standard MDP formulation, with emphasis on its boundedness and non-negativity. We further note our use of notation $S_t^{\boldsymbol{\pi}}$ for the (random) state visited at time $t$ on a trajectory generated by following policy $\boldsymbol{\pi}$ and initialized at $S_\tau = s_\tau$.

Next, we define action-value or Q-function that relates to the value function in (5).

**Definition 3.6** (Q-function). For each state-action pair $(s,a) \in \mathcal{S} \times \mathcal{A}$ and a fixed policy $\boldsymbol{\pi} \in \Pi^{MD}$, we define *Q-function* as

$$Q^{\boldsymbol{\pi}}(s,a) \doteq \mathbb{E}\left[\sum_{t=0}^{\infty} \varphi(t)R(S_t^{a\cdot\boldsymbol{\pi}}, \boldsymbol{\pi}(S_t^{a\cdot\boldsymbol{\pi}}))\Big| S_0 = s\right] \tag{6}$$

Note that to emphasize on the stationarity of our problem, we have avoided any explicit appearance of $\tau$ in the Definition 3.6 above by performing a simple change of variable drawn from the Markov property $\mathbb{P}[S_t^{\boldsymbol{\pi}} = s_t | S_\tau = s_\tau] = \mathbb{P}[S_{t-\tau}^{\boldsymbol{\pi}} = s_t | S_0 = s_\tau]$. Later in Section 4.2, $\tau$ will be re-introduced as our RL agent's parameter that keeps track of nonstationarity changes.

We may now revisit the TIC concept by witnessing how criterion (5) violates BPO through a Gridworld counterexample.

*Example* 3.7 (BPO Violation). Consider a Gridworld environment as described in Figure 1(a) and a hyperbolically-discounted criterion, i.e., setting $\varphi(t-\tau) = 1/(1 + k(t-\tau))$ in (5). Given $s_0 = 21$, we can compute (by trajectory enumeration) the globally-optimal, precommitment policy $\boldsymbol{\pi}^{*0}$ as in Figure 1(b). After applying delay $\tau = 3$ and following the delaying policy $\boldsymbol{\delta}^\tau = \boldsymbol{\pi}^{*0}$, we reach $s_\tau = 9$, at which the locally-optimal policy suggests $\boldsymbol{\pi}^{*\tau} \doteq \{\pi^{*\tau}(9)\} = \{\leftarrow\}$ and accrues rewards $V^{\boldsymbol{\pi}^{*\tau}}(s_\tau) = 10/(1+1) > 19/(1+3) = V^{\boldsymbol{\pi}^{*0}}(s_\tau)$; see Figure 1(d). This violates BPO at $\tau = 3$ and $s_\tau = 9$.

Here onwards, we will use the general-discounting TIC value function and Q-function as defined in (5)-(6) for any appearance of $V^{\boldsymbol{\pi}}(s)$ and $Q^{\boldsymbol{\pi}}(s,a)$, unless specified otherwise. We then denote by $V_{TC}^{\boldsymbol{\pi}}(s)$ and $Q_{TC}^{\boldsymbol{\pi}}(s,a)$ the exponential-discounting (i.e., $\varphi(t) = \gamma^t$) time-consistent (TC) value function and Q-function to exemplify any standard RL formulations in the subsequent sections.

## 4 An Analysis of Standard PI

In this section, we analyze standard PI under the SPE optimality type, revealing its merits and insufficiencies.

### 4.1 SPE Optimality of Termination Policy

We first present the standard PI update,

$$\pi'(s) \leftarrow \arg\max_{a \in \mathcal{A}_s} Q^{\boldsymbol{\pi}}(s,a), \forall s \in \mathcal{S} \tag{7}$$

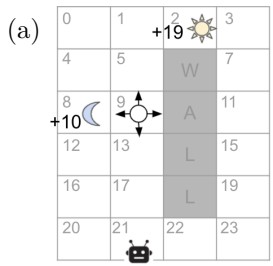

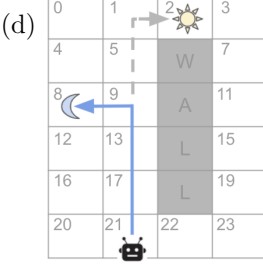

Figure 1: **(a) Deterministic, Hyperbolic ($k=1$) Gridworld.** $\mathcal{S}$ comprises 2 absorbing states $\bar{S} = \{\bar{2}, \bar{8}\}$ emitting rewards $R(\bar{2}) = 19, R(\bar{8}) = 10$. Each action in $\mathcal{A} = \{\uparrow, \rightarrow, \downarrow, \leftarrow\}$ drives transition through deterministic $P$; transitions to WALL or outside the grid will spawn the agent back to its original location. **(b) Globally-optimal, precommitment policy $\boldsymbol{\pi}^{*0}$** and its corresponding path, accruing accumulated rewards $V^{\boldsymbol{\pi}^{*0}}(s_0) = \frac{19}{1+6}$. This path exhibits TIC at $\tau = 3$ and $s_\tau = 9$ as shown in Example 3.7. **(c) SPE policy $\hat{\boldsymbol{\pi}}$** and its corresponding path, accruing rewards $V^{\hat{\boldsymbol{\pi}}}(s_0) = \frac{19}{1+8} < V^{\boldsymbol{\pi}^{*0}}(s_0)$. One could refer back to Definition 3.1 and verify that no states $s_\tau$ on this path have the incentive to deviate from its current policy $\hat{\pi}(s_\tau)$. **(d) Delusional policy $\boldsymbol{\delta}^\tau \cdot \boldsymbol{\pi}^{*\tau}$** and its corresponding path, with $\tau, \boldsymbol{\delta}^\tau$, and $\boldsymbol{\pi}^{*\tau}$ specified in Example 3.7, accruing rewards of $V^{\boldsymbol{\delta}^\tau \cdot \boldsymbol{\pi}^{*\tau}}(s_0) = \frac{10}{1+4} < V^{\hat{\boldsymbol{\pi}}}(s_0)$. The term 'delusional' is used to reflect how state 21 presumes 9 will go up, unaware of the TIC issue.

where $\boldsymbol{\pi}', \boldsymbol{\pi}$ represent *new* (at *current* iteration) and *old* policies (at *previous* iteration) in any two consecutive iterations. Next, we will show the merit of standard PI in Proposition 4.1: its termination policy achieves SPE optimality.

**Proposition 4.1.** *If $\boldsymbol{\pi}' = \boldsymbol{\pi}$ and update follows the rule in (7), then $\boldsymbol{\pi}, \boldsymbol{\pi}'$ are SPE policy.*

*Proof.* By (7) and $\boldsymbol{\pi}' = \boldsymbol{\pi}$, we obtain that $\forall a \in \mathcal{A}_s, \forall s \in \mathcal{S}$,

$$Q^{\boldsymbol{\pi}}(s, \pi'(s)) \geq Q^{\boldsymbol{\pi}}(s, a) \quad \Rightarrow \quad Q^{\boldsymbol{\pi}}(s, \pi(s)) \geq Q^{\boldsymbol{\pi}}(s, a).$$

Thus, by Definition 3.1, $\boldsymbol{\pi}, \boldsymbol{\pi}'$ are SPE policy. □

### 4.2 Policy Evaluation

The update rule (7) requires the computation of the true TIC Q-function $Q^{\boldsymbol{\pi}}(s, a)$, which is not straightforward. In standard RL setting, there is a DP formula to efficiently compute TC Q-function,

$$Q_{TC}^{\boldsymbol{\pi}}(s_t, a_t) = \mathbb{E}_{R, S' \sim p_{s_t}^{a_t}}[R(s_t, a_t) + \gamma V_{TC}^{\boldsymbol{\pi}}(S')], \tag{8}$$

where $V_{TC}^{\boldsymbol{\pi}}(s)$ (iteratively) solves (8) after substituting $\pi(s_t)$ into $a_t$. Under general discounting as in (5), (8) no longer holds. In this subsection, we present a recursive formula satisfied by our TIC Q-function (see (13) below) by leveraging the extended DP theory (Björk et al. (2014)).

**TIC-adjusted DP** Noting that in Section 3.2 we have assumed access to a fixed $\bar{T}_\epsilon < \infty$, we introduce a reward adjustment (or $r$-function) that our agent will use it to track the nonstationary changes (due to TIC) in Q-function.

**Definition 4.2** ($r$-function)**.** For each $\tau \in \{0, \dots, \bar{T}_\epsilon\}$, $m \in \{\tau, \dots, \bar{T}_\epsilon\}$, $s \in \mathcal{S}$, $a \in \mathcal{A}$, and a fixed policy $\boldsymbol{\pi} \in \Pi^{MD}$, we define $r$-function as

$$r^{\boldsymbol{\pi},\tau,m}(s,a) \doteq \mathbb{E}\left[\varphi(m-\tau)R\left(S_m^{a\cdot\boldsymbol{\pi}}, \pi(S_m^{a\cdot\boldsymbol{\pi}})\right)|S_\tau = s\right] \tag{9}$$

where $\tau$ and $m$ are fixed parameters.

Next, we will use the adjustment function above to obtain a formula that recursively computes our Q-function.

**Theorem 4.3.** *For any fixed* $\boldsymbol{\pi} \in \Pi^{MD}, \tau \in \{0, \dots, \bar{T}_\epsilon\}, m \in \{\tau, \dots, \bar{T}_\epsilon\}, s \in \mathcal{S}, a \in \mathcal{A}_s$, *$r$-function satisfies for $m = \tau$,*

$$r^{\boldsymbol{\pi},\tau,\tau}(s,a) = \mathbb{E}_{R \sim p_s^a}\left[\varphi(0)R(s,a)\right] \tag{10}$$

*and for $m \geq \tau + 1$,*

$$r^{\boldsymbol{\pi},\tau,m}(s,a) = \mathbb{E}_{S' \sim p_s^a}\left[\frac{\varphi(m-\tau)}{\varphi(m-(\tau+1))} r^{\boldsymbol{\pi},\tau+1,m}(S', \pi(S'))\right]. \tag{11}$$

*Moreover, for any fixed* $\boldsymbol{\pi} \in \Pi^{MD}, t \in \{0, \dots, \bar{T}_\epsilon\}, s_t \in \mathcal{S}, a_t \in \mathcal{A}_{s_t}$, *under some technical conditions (see Assumption B.1), Q-function satisfies for $s_t \in \bar{\mathcal{S}}$ and $a_t \in \bar{\mathcal{A}}$,*

$$Q^{\boldsymbol{\pi}}(s_t, a_t) = \mathbb{E}_{R \sim p_{s_t}^{a_t}}\left[\varphi(0)R(s_t, a_t)\right] \tag{12}$$

*and for $s_t \in \mathcal{S} \setminus \bar{\mathcal{S}}$ and $a_t \in \mathcal{A}_{s_t}$,*

$$Q^{\boldsymbol{\pi}}(s_t, a_t) \approx \mathbb{E}_{R \sim p_{s_t}^{a_t}}[\varphi(0)R(s_t, a_t)] + \mathbb{E}_{S' \sim p_{s_t}^{a_t}}[Q^{\boldsymbol{\pi}}(S', \pi(S'))] - \Delta r_t^{\boldsymbol{\pi}}, \tag{13}$$

*where* $\Delta r_t^{\boldsymbol{\pi}} \doteq \sum_{m=t+1}^{\bar{T}_\epsilon} \left(\mathbb{E}_{S' \sim p_{s_t}^{a_t}}\left[r^{\boldsymbol{\pi},t+1,m}(S', \pi(S'))\right] - r^{\boldsymbol{\pi},t,m}(s_t, a_t)\right)$.

*Remark* 4.4. Both the proof of Theorem 4.3 and the technical conditions for it are provided in Appendix B. We note that Theorem 4.3 is an analog to Proposition 11 in Lesmana & Pun (2021) and our main technical novelty lies in the approximation ('$\approx$') part of (13); here, $a \approx b$ denotes the existence of $\kappa(\epsilon)$ s.t. $|a - b| \leq \frac{\epsilon}{\kappa(\epsilon)}$ and $\lim_{\epsilon \downarrow 0} \frac{\epsilon}{\kappa(\epsilon)} \downarrow 0$. Intuitively, this result ensures that our approximation error can be made arbitrarily small by choosing a sufficiently large $\bar{T}_\epsilon$.

*Remark* 4.5. Theorem 4.3 has used the specific properties of general-discounting TIC source. For other types of TIC sources, recursive formulas need to be re-derived. In risk-sensitive case, for instance, readers may refer to Tamar & Mannor (2013); Sobel (1982).

Standard TD-based approximation algorithms such as Q-learning Watkins & Dayan (1992) are drawn from the standard formula (8) and thus, are insufficient for general-discounting factor. Theorem 4.3 provides a formula that addresses insufficiency of standard formula (8), which we will later use to reinvent a new approximate PI algorithm for general-discounting objectives.

### 4.3 Policy Improvement (Update Monotonicity)

In this subsection, we will highlight some insufficiencies of the standard PI's update and analysis tools in the face of TIC by revisiting the unprovable policy improvement theorem. To start off, we present the following proof (cf. Sutton & Barto (2018)) of the theorem: $\forall s \in \mathcal{S}$,

$$\begin{aligned} V_{TC}^{\boldsymbol{\pi}}(s) &\leq Q_{TC}^{\boldsymbol{\pi}}(s, \pi'(s)) = \mathbb{E}\left[R_{t+1}^{\pi'} + \gamma V_{TC}^{\boldsymbol{\pi}}(S_{t+1}^{\pi'})|S_t = s\right] \\ &\leq \mathbb{E}\left[R_{t+1}^{\pi'} + \gamma Q_{TC}^{\boldsymbol{\pi}}(S_{t+1}^{\pi'}, \pi'(S_{t+1}^{\pi'}))|S_t = s\right] = \cdots \leq \cdots \\ &\leq V_{TC}^{\pi'}(s). \end{aligned} \tag{14}$$

Note that in each alternating step of '=' and '≤', two operations are performed: (i) a recursive expansion of TC Q-function, and (ii) substituting the **monotonicity** relation:

$$\forall s \in \mathcal{S}, \ V_{TC}^{\pi' \cdot \boldsymbol{\pi}}(s) \geq V_{TC}^{\pi \cdot \boldsymbol{\pi}}(s) \ \Rightarrow \forall s \in \mathcal{S}, \ V_{TC}^{\boldsymbol{\delta}^\tau \cdot \pi' \cdot \boldsymbol{\pi}}(s) \geq V_{TC}^{\boldsymbol{\delta}^\tau \cdot \pi \cdot \boldsymbol{\pi}}(s) \tag{15}$$

for all delays $\tau \geq 1$ and $\boldsymbol{\delta}^\tau = \{\pi', \pi', \dots\}$. Let us pay attention to the monotonicity relation, particularly about how (15) fails under a TIC criterion. To this end, we recall Example 3.7 and focus on the states along the precommitment path in Figure 1(b). We can then counter (15) as follows:

Set $\boldsymbol{\pi} = \boldsymbol{\pi}^{*0}, \tau = 3, \boldsymbol{\delta}^\tau = \boldsymbol{\pi}^{*0}, \pi'(9) = \leftarrow$; then, $\frac{19}{1+3} = V^{\pi \cdot \boldsymbol{\pi}}(9) \leq V^{\pi' \cdot \boldsymbol{\pi}}(9) = \frac{10}{1+1}$ holds. However,

$$\frac{19}{1+6} = V^{\boldsymbol{\delta}^3 \cdot \pi \cdot \boldsymbol{\pi}}(21) = \mathbb{E}_{\boldsymbol{\delta}^3}[V^{\pi \cdot \boldsymbol{\pi}}(9)] > \mathbb{E}_{\boldsymbol{\delta}^3}[V^{\pi' \cdot \boldsymbol{\pi}}(9)] = V^{\boldsymbol{\delta}^3 \cdot \pi' \cdot \boldsymbol{\pi}}(21) = \frac{10}{1+4} \tag{16}$$

showing that at $s = 21$, the monotonicity relation (15) does not hold.

We make two observations here: (i) defining improvement as in policy improvement theorem (i.e., $\forall s \in \mathcal{S}, V^{\boldsymbol{\pi}'}(s) \geq V^{\boldsymbol{\pi}}(s)$) might be too strong as this definition targets optimal policies *not* SPE policies, (ii) the counterexample (16) suggests the existence of *priority ordering*[4] over $\mathcal{S}$ (i.e. 9 holds *priority* over 21) such that unordered (i.e. $\forall s \in \mathcal{S}$) update as in (7) may not suffice. To further probe on these issues, we will consider the following example.

*Example* 4.6 (Inefficiency of Standard PI). Let us refer back our Hyperbolic Gridworld in Figure 1(a). We will keep our deterministic transition and reward functions, but restrict our state-space to:

$$\tilde{\mathcal{S}} = \{1, 2, 3, 5, 7, 8, 9, 11, 13, 15, 17, 19, 21, 22, 23\}$$

and action-spaces to:

$$\begin{aligned}
\tilde{\mathcal{A}}_{21} &= \{\uparrow, \rightarrow\}, \quad \tilde{\mathcal{A}}_9 = \{\uparrow, \leftarrow\}, \\
\mathcal{A}_3 &= \{\leftarrow\}, \\
\mathcal{A}_s &= \{\rightarrow\}, \quad \forall s \in \{1, 22\}, \\
\mathcal{A}_s &= \{\uparrow\}, \quad \forall s \in \{17, 13, 5, 7, 11, 15, 19, 23\}, \\
\mathcal{A}_s &= \{\bar{a}\}, \quad \forall s \in \{2, 8\}.
\end{aligned}$$

Letting $s_0 = 21, \epsilon = 0, \tilde{\Pi} = \{\boldsymbol{\pi} \in \Pi^{MD} | \boldsymbol{\pi} : \tilde{\mathcal{S}} \to \tilde{\mathcal{A}}_s\}$, we have a *priority-ordering* on $\tilde{\mathcal{S}}$, $\tilde{\mathcal{S}}_{0:\bar{T}} = \tilde{\mathcal{S}}_{0:\bar{T}_0}^{s_0, \tilde{\Pi}}$:

$$\tilde{\mathcal{S}}_0 = \{21\}, \ \tilde{\mathcal{S}}_1 = \{17, 22\}, \ \tilde{\mathcal{S}}_2 = \{13, 23\}, \ \tilde{\mathcal{S}}_3 = \{9, 19\}, \ \tilde{\mathcal{S}}_4 = \{5, 8, 15\},$$

$$\tilde{\mathcal{S}}_5 = \{1, 11\}, \ \tilde{\mathcal{S}}_6 = \{2, 7\}, \ \tilde{\mathcal{S}}_7 = \{3\}, \ \tilde{\mathcal{S}}_8 = \{2\}. \tag{17}$$

To apply standard PI for SPE policy search from $s_0 = 21$, we can choose an initial policy $\boldsymbol{\pi}^{(0)}$ as illustrated in Figure 2(a). Following the standard PI's rule (7), policies at *all states* are updated conditional to the policy at *previous* iteration. Let us now focus on the two important states $s = 9, 21$, in which decisions need to be made (i.e., $|\tilde{\mathcal{A}}_s| > 1$). First, note that after updated conditional to the *previous* policy $\pi^{(0)}(9) = \uparrow$, we obtain $\pi^{(1)}(21) = \uparrow$ that incurs a higher reward; see Figure 2(b). Following the same rule, we also obtain $\pi^{(1)}(9) = \leftarrow$. When combined, the *current* iteration ends up in $\boldsymbol{\pi}^{(1)}$ that corresponds to a *delusional* path; see Figure 2(c). We contend that such iterative update is inefficient as we would have found the desired SPE path if state 21 has anticipated $\pi^{(1)}(9) = \leftarrow$ when making its update. We will see how we can achieve this in Section 5.1, by leveraging a known *priority-ordering* as in (17).

# 5 Backward Q-learning Algorithm

Drawing on the analyses and observations in Section 4, we propose a new algorithm in the approximate PI family that targets SPE policy under a general-discounting criterion.

---

[4]By either the sophisticated agent's strategy Strotz (1955) or its SPE formalism Björk et al. (2014), *higher priority* here corresponds to a *later order of visitation* in a trajectory.

(a) 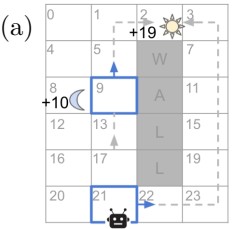 (b) 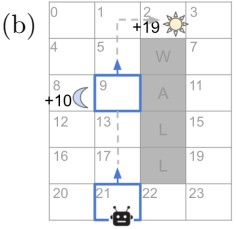 (c) 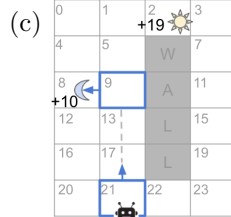

Figure 2: **2-Layered Correction with Standard PI.** (a) Policies at initialization. (b) Policies after state 21 updates. (c) Policies after state 9 (and all other states) updates.

## 5.1 Backward Conditioning

To mitigate the insufficiencies surrounding update monotonicity, we build on a recent result in Lesmana & Pun (2021) and propose *backward conditioning*: to perform update *backward* from $\mathcal{S}_{\bar{T}}$ to $\mathcal{S}_0$ and *conditioning* the update of states with *lower priority* (happens *earlier*) on the *new* policy $\boldsymbol{\pi}'$ of states with *higher priority* (happens *later*). We formalize the above in the following update rule.

**Definition 5.1** (*Backward Conditioning* Rule). For any $\epsilon > 0$, set $\bar{T} := \bar{T}_\epsilon$ and let $\mathcal{S}_{0:\bar{T}} := \mathcal{S}_{0:\bar{T}}^{s_0,\Pi^{MD}}$ be a *priority-ordering* on $\mathcal{S}$. Then, for $t = \bar{T} - 1 : 0$:

$$\forall s \in \mathcal{S}_t, \pi'(s) \leftarrow \arg\max_{a \in \mathcal{A}_s} Q^{(\boldsymbol{\pi}')^{\bar{T}-1-t} \cdot \boldsymbol{\pi}}(s, a) \tag{18}$$

*Remark* 5.2. Note that in the Definition 5.1, we have assumed the existence of a *priority-ordering*: whatever actions the states in $\mathcal{S}_{0:t-1}$ are taking are assumed to have no effect on the choice of states in $\mathcal{S}_t$. This justifies (18): its *backward* order and conditioning the update of any $s \in \mathcal{S}_{\bar{T}-1}$ (with *highest* priority) on the *old* policy $\boldsymbol{\pi}$. We note however that even without such *priority-ordering*, the worst that can happen is $\mathcal{S}_t^{s_0,\Pi^{MD}} = \mathcal{S}, \forall t \in [0, \bar{T}]$, which is equivalent to performing standard PI in (7) $\bar{T}$ times.

Next, we will show that the backward conditioning rule preserves the SPE optimality of termination policy.

**Proposition 5.3.** *If* $\boldsymbol{\pi}' = \boldsymbol{\pi}$ *and update follows the rule in equation 18, then* $\boldsymbol{\pi}, \boldsymbol{\pi}'$ *are SPE policy.*

*Proof.* First, we will show that

$$\forall s \in \mathcal{S}, s \in \mathcal{S}_{0:\bar{T}-1}^{s_0,\Pi^{MD}} \tag{19}$$

Since $\bar{T}_\epsilon - 1 \geq \hat{T}_{s_0}$, by definition of $\hat{T}_{s_0}$, $\forall s \in \mathcal{S}$,

$$\exists \boldsymbol{\pi} \in \Pi^{MD}, \sum_{t=0}^{\bar{T}_\epsilon - 1} \mathbb{P}[S_t^{\boldsymbol{\pi}} = s|s_0] > 0 \Rightarrow \exists t \in [0, \bar{T}_\epsilon - 1] \text{ s.t. } \exists \boldsymbol{\pi} \in \Pi^{MD}, \mathbb{P}[S_t^{\boldsymbol{\pi}} = s|s_0] > 0$$

$$\Rightarrow \exists t \in [0, \bar{T}_\epsilon - 1] \text{ s.t. } s \in \mathcal{S}_t$$

$$\Rightarrow s \in \mathcal{S}_{0:\bar{T}_\epsilon - 1}.$$

Now, let us consider arbitrary $s \in \mathcal{S}$. By (19), $\exists t \in [0, \bar{T} - 1]$ s.t. $s \in \mathcal{S}_t$. Using such $t$, by (18), we have $\forall a \in \mathcal{A}_s$,

$$Q^{(\boldsymbol{\pi}')^{\bar{T}-t-1} \cdot \boldsymbol{\pi}}(s, \pi'(s)) \geq Q^{(\boldsymbol{\pi}')^{\bar{T}-t-1} \cdot \boldsymbol{\pi}}(s, a)$$

which by $\boldsymbol{\pi} = \boldsymbol{\pi}'$, implies

$$Q^{\boldsymbol{\pi}'}(s, \pi'(s)) \geq Q^{\boldsymbol{\pi}'}(s, a).$$

$\square$

*Example* 5.4 (Efficiency of Backward Conditioning). To illustrate the difference between (18) and (7), let us reconsider the setup in Example 4.6. Given the same initialization (see Figure 3(a)), backward conditioning rule (18) asserts state $9 \in \tilde{\mathcal{S}}_3$ to update *earlier* than state $21 \in \tilde{\mathcal{S}}_0$, in one *current* iteration. This necessarily means that by the time state 21 updates, it will have anticipated state 9's *current* policy $\pi^{(1)}(9) = \leftarrow$ (see Figure 3(b)) and directly obtain the desired $\pi^{(1)}(21) = \rightarrow$ (see Figure 3(c)). When combined, such *current* iteration ends up in $\boldsymbol{\pi}^{(1)}$ that corresponds to the target SPE path $\hat{\boldsymbol{\pi}}$; note how the path extending from $s0 = 21$ in Figure 3(c) overlaps with the one in Figure 1(c). In contrast to (7), backward conditioning imposes that the choice of *later states* are directly propagated to *earlier states* in each policy iteration and correspondingly prevents inefficient movement of policies (i.e., away from an SPE policy as in Figure 2).

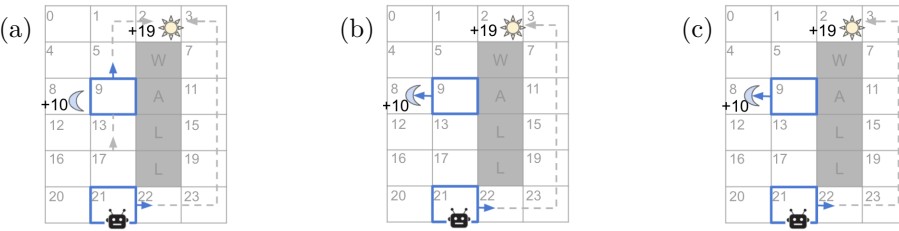

Figure 3: **2-Layered Correction with Backward Conditioning.** (a) Initialization. (b) State 9 updates first. (c) State 21 updates last, after $9, 13, 17$ make their updates.

*Remark* 5.5. Through Example 4.6 and Example 5.4, we have concluded that relative to standard PI, backward conditioning mitigates inefficient movement of policies (cf. Figure 2) by propagating information about future players' "optimal" policies (as soon as it is found) directly to earlier players (cf. Figure 3). To see how this phenomenon transfers to a TIC "Q-learning" setting, readers may refer to Table 1 in Section 6.1 and Appendix D.4.3, particularly to (i) the reduced delusionality in earlier episodes under backward conditioning (cf. Figure 6(e)) as compared to standard forwardly-ordered version (cf. Figure 9(e)), and (ii) the improved stability of Figures 6(e)-(f) relative to Figures 9(e)-(f). We would like to remark that our observations are consistent with the results of related literature under non-TIC motivations, e.g., sample efficiency (Lin (1991); Lee et al. (2019)), consistent uncertainty propagation (Bai et al. (2021)), where they have similarly concluded backward update's power in information propagation.

*Remark* 5.6. In Theorem 25 of Lesmana & Pun (2021), a finite-horizon analog to the update rule (18) has been shown to exhibit lex-monotonicity (i.e. a weaker update monotonicity than PIT that reflects *closer to SPE*), by leveraging a policy-independent ordering on time-extended state-space due to $T < \infty$ (i.e. players are times). This prevents the cycling of policies, implying convergence. In $T = \infty$, we lose this order (i.e. *players are states*) and resort to using *visitation order* on a trajectory. This results in a lex-mono analog: $\forall t$, if $\boldsymbol{\pi}'_{\mathcal{S}_{>t}}$ is *closer to SPE* $\hat{\boldsymbol{\pi}}_{\mathcal{S}_{>t}}$ than $\boldsymbol{\pi}_{\mathcal{S}_{>t}}$; so is $\pi'_{\mathcal{S}_t}$. It was discussed through Example 5.4 when $\mathcal{S}_t = \{21\}$ and $\tilde{\mathcal{S}}_{>t} = \{17, 13, 9\}$. Convergence thus remains open as complications may arise when $\mathcal{S}_t \cap \mathcal{S}_{t'} \neq \emptyset$ for some $t \neq t'$.

## 5.2 Approximate Backward Conditioning

In this subsection, we are interested in deriving an approximate version to the backward conditioning rule in Definition 5.1. This can be done in two steps. Firstly, we will replace the exact computation of $Q^{\boldsymbol{\pi}'}(s, a)$ with *prediction*. To this end, we will use our results in Theorem 4.3 and derive TIC-adjusted TD targets for predicting $r^{\boldsymbol{\pi}'}(s_t, a_t)$ from (10)-(11) and $Q^{\boldsymbol{\pi}'}(s_t, a_t)$ from (12)-(13),

$$
\xi_t^r(m) = \begin{cases} \varphi(0) R(s_t, a_t), & m = t \\ \frac{\gamma(m-t)}{\gamma(m-(t+1))} r^{t+1,m}(S_{t+1}, \pi'(S_{t+1})), & m > t \end{cases} \tag{20}
$$

$$\xi_t^Q = \begin{cases} \varphi(0)R(s_t, a_t) + Q(S_{t+1}, \pi'(S_{t+1})) \\ -\max(0, \Delta r_t), \quad t \leq T^* - 1 \\ \varphi(0)R(s_t, a_t), \quad t = T^* \text{ and } s_t \in \bar{\mathcal{S}} \end{cases} \tag{21}$$

where $\Delta r_t = \sum_{m=t+1}^{T^*} r^{t+1,m}(s_{t+1}, \pi'(s_{t+1})) - r^{t,m}(s_t, a_t)$. Then, we can incorporate the backward conditioning simply by reordering update from the end of a sampled trajectory $t = T^*$ to $t = 0$. Note that here, we have replaced $\bar{T}$ with $T^* \leq \bar{T}$ to account for the variable length of trajectories encountered in practice. We summarize this section with a pseudocode in Algorithm 1, where lines $11, 18 - 20$ capture our backward conditioning and lines $12 - 17$ capture the *TIC-adjusted TD* evaluation[5].

---

**Algorithm 1** Backward Q-learning (bwdQ)

---

1: **Parameters:** exploration rate $\epsilon$, episode length $\bar{T}$, learning rates $\alpha_Q, \alpha_r$
2: **Init:**
3: $Q(s, a) = 0, \forall s \in \mathcal{S} \setminus \bar{\mathcal{S}}, a \in \mathcal{A}$;
4: $Q(s, a) = \varphi(0)R(s, a), \forall s \in \bar{\mathcal{S}}, a \in \bar{\mathcal{A}}$;
5: $r^{\tau,m}(s, a) = 0, \forall \tau, m, s \in \mathcal{S}, a \in \mathcal{A}$;
6: $\pi'(s) \leftarrow \arg\max_a Q(s, a), \forall s \in \mathcal{S}, \boldsymbol{\pi} \leftarrow \emptyset$
7: **repeat**
8:     $\boldsymbol{\pi} \leftarrow \boldsymbol{\pi}'$;
9:     Choose $S_0$ randomly;
10:     Sample $S_0, A_0, \ldots, S_{T^*-1}, A_{T^*-1}, S_{T^*}, A_{T^*} = \bar{a} \sim \boldsymbol{\pi}^\epsilon$;
11:     **for** $t \leftarrow T^*$ **to** 0 **do**
12:       **for** $m \leftarrow t$ **to** $T^*$ **do**
13:         Compute $\xi_t^r(m)$ according to (20);
14:         $r^{t,m}(S_t, A_t) \leftarrow r^{t,m}(S_t, A_t) + \alpha_r(\xi_t^r(m) - r^{t,m}(S_t, A_t))$;
15:       **end for**
16:       Compute $\xi_t^Q$ according to (21);
17:       $Q(S_t, A_t) \leftarrow Q(S_t, A_t) + \alpha_Q(\xi_t^Q - Q(S_t, A_t))$;
18:       **if** $Q(S_t, \pi(S_t)) < \max_a Q(S_t, a)$ **then**
19:         $\pi'(S_t) \leftarrow \arg\max_a Q(S_t, a)$
20:       **else**
21:         $\pi'(S_t) \leftarrow \pi(S_t)$
22:       **end if**
23:     **end for**
24: **until** stable ($\boldsymbol{\pi}' \neq \boldsymbol{\pi}$)

---

*Remark* 5.7. While Algorithm 1 can be considered as a Q-learning's variant, standard convergence analysis such as in Bertsekas & Tsitsiklis (1996) does not apply to our case. Even more recent analysis techniques on coupled iterations such as those done for double Q-learning (Hasselt (2010); Xiong et al. (2020); Zhao et al. (2021)) do not apply to our case for the fully-coupled dynamics of the iterated $r$-functions and $Q$-function. Formal convergence analysis of backward Q-learning is thus left for future study.

## 6 Learning Performance: An Illustration

In this section, we illustrate the behaviour of bwdQ in two TIC Gridworld environments: (i) **Deterministic (D)**, by reusing our motivating example in Figure 1, which has been shown to exhibit future deviations, and (ii) **Stochastic (S)**, by injecting some random noise into state 9's transition in (D). For our comparative study, we consider as benchmarks two approximate PI variants that also target SPE policy under general-discounting objectives, namely *standard PI with Monte Carlo (MC)* and *sophisticated EU (sophEU)* from Evans et al. (2016). Pseudocodes and training specifications are provided in Appendices D.2-D.3.

---

[5]For detailed derivations of Algorithm 1, readers can refer to Appendix C.

### 6.1 Results

Our results and evaluation can be segregated into three components: efficiency, value prediction, and termination policy, all of which are summarized into Table 1 and Figure 4.

**Efficiency**    In Section 5.1, we have provided an intuition on the desirability of *backward conditioning*. From Table 1, we can see its implication to actual learning instances with approximation. In particular, we can observe that bwdQ demonstrates higher learning efficiency in both (D) and (S): it has significantly shorter $\Delta i^*$ in average (mean) with lower standard deviation compared to the others.

Table 1: **Delusional period** $\Delta i^* \doteq |i^*_{21} - i^*_9|$ **statistics, presented as mean$_{\textbf{(stdev)}}$ (in thousands).** This metric relates to the 2-layered correction illustrated in Figures 2 and 3: $\Delta i^*$ quantifies how many iterations 21 needs to reflect 9's move to SPE. Episode indexes $i^*_9$ and $i^*_{21}$ represent the first overtaking episodes of mean SPE Q-value at states 9 and 21, respectively; see Appendix D.4.1 for illustrative Q-value curves. For each algorithm and environment, 10 experiments are conducted and each consists of 50 random seeds.

|       | MC | SophEU | BwdQ |
|-------|----|--------|------|
| (D)   | $15.39_{(3.69)}$ | $69.97_{(1.81)}$ | $2.37_{(0.73)}$ |
| (S)   | $14.55_{(4.83)}$ | $97.56_{(2.17)}$ | $3.68_{(0.51)}$ |

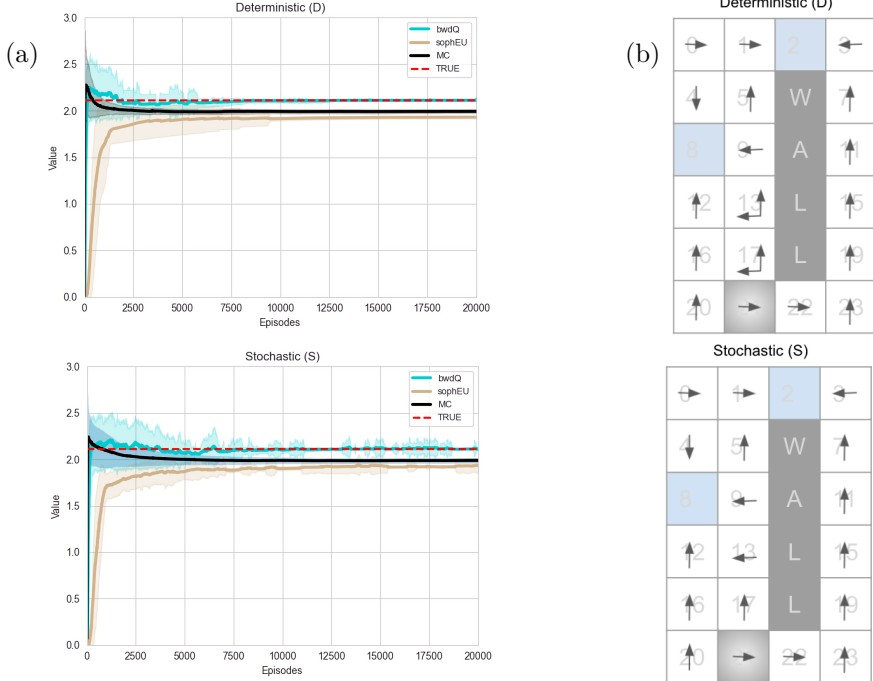

Figure 4: **(a) Value learning curves of** $s_0 = 21$. **(b) Termination policies**. For each algorithm in (a), the dark line and the shaded region each refers to the mean and standard deviation of 50 experiments, where in one experiment, we record the values $V^i(s_0) = Q^i(s_0, \pi^i(s_0))$ at the end of each training episode $i$. In (b), we record the termination policies $\pi^I(s), \forall s \in \mathcal{S}$ after the policies at all states stabilize at training episode $I$; note that $I$ here may differ across algorithms and experiments as we only want to show the asymptotic performance of each algorithm in learning SPE policy. Groundtruth 'TRUE' in (a) is then computed as the value of analytical SPE policy in (D), and the value of termination policy in (S).

**Value prediction**  From Figure 4(a), we can observe that in (D), the mean value of bwdQ matches closely the groundtruth (manually computed) upon convergence. On the contrary, sophEU and MC both converge at a value strictly smaller than the groundtruth. Similar conclusion can be drawn in (S), despite bwdQ produces higher variance than the rest; see Appendices D.4.2-D.4.3 for more results and discussions on value biases.

**Termination policy**  In both (D) and (S), *all algorithms* (i.e. MC, sophEU, and bwdQ) converge and the termination policies are plotted in Figure 4(b). While all algorithms converge to the same policies in (S), this is not true in (D): at $s = 13, 17$, MC and sophEU converge to $\{\uparrow, \uparrow\}$ when bwdQ converges to $\{\uparrow, \leftarrow\}$ or $\{\leftarrow, \leftarrow\}$. Thus, in Figure 4(b), we present together these three different termination policies. For the termination policies in (D), we can verify that they correspond to the groundtruth SPE policies (by Definition 3.1 and $Q^{\hat{\pi}}$ computed manually from the reward specifications in Figure 1). This is consistent with our results in Propositions 4.1 (MC) and 5.3 (bwdQ) that guarantee SPE optimality if converged. For the termination policies in (S), we can see how the noise injected to 9 affects the SPE policy: $\hat{\pi}(13)$ shifts from $\{\leftarrow, \uparrow\}$ in (D) to $\{\leftarrow\}$ in (S) as $Q^{\hat{\pi}}(13, \uparrow)$ in (S) is pulled down by random transitions of $9 \to 5$ and $9 \to 13$.

# 7   Conclusion and Future Works

Prior to this paper, it was unclear how policy iteration performs and whether it is sufficient in TIC RL settings under which BPO or DP becomes inaccessible. Through this paper, we demonstrated how introducing SPE optimality can shed lights on the two fundamental questions surrounding the use of PI in TIC RL setting. While this paper on TIC RL is of theoretical nature, we managed to use a toy Gridworld example to demonstrate our findings. In particular, we obtain positive results on PI in the sense of both standard PI and backward conditioning's capability to characterize SPE policies. Though we could not close the convergence of either standard PI or backward conditioning, we have made progress towards it by verifying the importance of *ordered* policy iteration and improvement criteria. From the perspective of policy evaluation, SPE optimality recovers the use of DP-like formulas. This has resulted in familiar forms of algorithms, such as our backward Q-learning, that is also important towards closing the analysis of SPE policy search.

From our current paper, closing the convergence analysis of PI, backward conditioning, and their approximate variants under SPE optimality (i.e., in particular, addressing the fully-coupled iterations that arise with TIC adjustments) is an important future direction. Secondly, noting our recovery of DP-like formulas and Q-learning like algorithms, we see promise in adopting the use of function approximations and scaling up to more general models such as linear MDP (Jin et al. (2020)) or more complex domains such as Atari (Bellemare et al. (2013)). Finally, it will be interesting to apply the analysis of this paper to other TIC sources such as risk-sensitive objectives and compare across different TIC sources the extent of future deviations, TIC-adjustment techniques, and the control performance of PI (as a proxy for SPE policy). On a broader extent, we hope that our first attempt can inspire more works on TIC RL towards closing the gap between progress in standard RL and TIC RL.

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

## A  Additional Details on Assumption 3.3-3.5

### A.1  MDP Examples under General-Discounting Criterion

In this section, we derive several sufficient conditions for our assumptions in Section 3.2, in the form of restrictions on MDP $(\mathcal{S}, \mathcal{A}, \mathbb{P}, R)$ or discounting function $\varphi(\cdot)$.

**Definition A.1** (*Boundary-only* Rewards)**.** The reward function $R : \mathcal{S} \times \mathcal{A} \to \mathbb{R}^+$ is *boundary-only* if it is non-zero only at *boundary* states, i.e. $R(s, a) > 0$ only if $s \in \bar{\mathcal{S}}$.

**Lemma A.2.** *Any MDP that has boundary-only rewards satisfies Assumption 3.3.*

*Proof.* Let $T_{\bar{\mathcal{S}}}^{\boldsymbol{\pi}}|s$ defines the *minimum* hitting time of any *boundary* states $\bar{s} \in \bar{\mathcal{S}}$ when initiated at $s$ and following $\boldsymbol{\pi}$. Thus, $\forall s \in \mathcal{S}, \forall \boldsymbol{\pi} \in \Pi^{MD}, \forall \bar{T} < \infty$,

$$V^{\boldsymbol{\pi}}(s) := \mathbb{E}\left[\sum_{t=0}^{\infty} \varphi(t) R_t^{\boldsymbol{\pi}} \mid s\right] = \sum_{\tau=0}^{\bar{T}} \mathbb{P}[T_{\bar{\mathcal{S}}}^{\boldsymbol{\pi}} = \tau|s]\mathbb{E}\left[\sum_{t=0}^{\tau} \varphi(t) R_t^{\boldsymbol{\pi}}|s\right] + \sum_{\tau=\bar{T}+1}^{\infty} \mathbb{P}[T_{\bar{\mathcal{S}}}^{\boldsymbol{\pi}} = \tau|s]\mathbb{E}\left[\sum_{t=0}^{\tau} \varphi(t) R_t^{\boldsymbol{\pi}}|s\right],$$

$$\begin{aligned}
V^{\boldsymbol{\pi}^{\bar{T}} \cdot \ddot{\boldsymbol{\pi}}}(s) &:= \mathbb{E}\left[\sum_{t=0}^{\infty} \varphi(t) R_t^{\boldsymbol{\pi}^{\bar{T}} \cdot \ddot{\boldsymbol{\pi}}} \mid s\right] \\
&= \sum_{\tau=0}^{\bar{T}} \mathbb{P}[T_{\bar{\mathcal{S}}}^{\boldsymbol{\pi}^{\bar{T}} \cdot \ddot{\boldsymbol{\pi}}} = \tau|s]\mathbb{E}\left[\sum_{t=0}^{\tau} \varphi(t) R_t^{\boldsymbol{\pi}^{\bar{T}} \cdot \ddot{\boldsymbol{\pi}}}|s\right] + \sum_{\tau=\bar{T}+1}^{\infty} \mathbb{P}[T_{\bar{\mathcal{S}}}^{\boldsymbol{\pi}^{\bar{T}} \cdot \ddot{\boldsymbol{\pi}}} = \tau|s]\mathbb{E}\left[\sum_{t=0}^{\tau} \varphi(t) R_t^{\boldsymbol{\pi}^{\bar{T}} \cdot \ddot{\boldsymbol{\pi}}}|s\right] \\
&= \sum_{\tau=0}^{\bar{T}} \mathbb{P}[T_{\bar{\mathcal{S}}}^{\boldsymbol{\pi}} = \tau|s]\mathbb{E}\left[\sum_{t=0}^{\tau} \varphi(t) R_t^{\boldsymbol{\pi}}|s\right]. \qquad (\text{since } \forall \tau \geq \bar{T}+1, S_\tau = \bar{s}_{\text{void}} \Rightarrow \mathbb{P}[T_{\bar{\mathcal{S}}}^{\boldsymbol{\pi}^{\bar{T}} \cdot \ddot{\boldsymbol{\pi}}} = \tau|s] = 0.)
\end{aligned}$$

By bounded reward function,

$$R_{max} := \max\{|R(s, a)| : s \in \mathcal{S}, a \in \mathcal{A}\} \tag{22}$$

exists. Then, $\forall \epsilon > 0, \forall s \in \mathcal{S}$, we can set $\bar{T} < \infty$ s.t.

$$|R_{\max}|\varphi(\bar{T} + 1) \leq \epsilon \tag{23}$$

and the following holds,

$$\begin{aligned}
\sup_{\boldsymbol{\pi} \in \Pi^{MD}} |V^{\boldsymbol{\pi}^{\bar{T}} \cdot \boldsymbol{\pi}}(s) - V^{\boldsymbol{\pi}^{\bar{T}} \cdot \ddot{\boldsymbol{\pi}}}(s)| &= \sup_{\boldsymbol{\pi} \in \Pi^{MD}} \left|\sum_{\tau=\bar{T}+1}^{\infty} \mathbb{P}[T_{\bar{\mathcal{S}}}^{\boldsymbol{\pi}} = \tau|s]\mathbb{E}\left[\varphi(\tau) R_\tau^{\boldsymbol{\pi}}|s\right]\right| && (\text{by } \textit{boundary-only} \text{ rewards}) \\
&\leq \sup_{\boldsymbol{\pi} \in \Pi^{MD}} |R_{\max}| \sum_{\tau=\bar{T}+1}^{\infty} \mathbb{P}[T_{\bar{\mathcal{S}}}^{\boldsymbol{\pi}} = \tau|s]\varphi(\tau) && (\text{by } (22)) \\
&\leq \sup_{\boldsymbol{\pi} \in \Pi^{MD}} |R_{\max}|\varphi(\bar{T} + 1)\mathbb{P}[T_{\bar{\mathcal{S}}}^{\boldsymbol{\pi}} > \bar{T}|s] && (\text{by } \varphi(\cdot) \text{ decreasing}) \\
&\leq |R_{\max}|\varphi(\bar{T} + 1) \leq \epsilon && (\text{by } (23))
\end{aligned}$$

$\qquad\qquad\qquad\qquad\qquad\qquad\qquad\qquad\qquad\qquad\qquad\qquad\qquad\qquad\qquad\qquad\qquad\qquad\qquad\qquad\qquad\qquad\qquad\qquad\qquad\qquad\qquad\square$

**Lemma A.3.** *Suppose an MDP has boundary-only rewards, $s_0$ that satisfies Assumption 3.4 such that $\bar{T}_{s_0,0} < \infty$, and a discounting factor $\varphi : \mathbb{N} \to (0, 1]$ that satisfies*

$$\forall t \geq 0, \frac{\varphi(\tau + t)}{\varphi(\tau)} \text{ is increasing in } \tau, \tau \geq 1. \tag{24}$$

*Then, Assumption 3.5 holds.*

*Proof.* Suppose otherwise, $\exists \epsilon^* > 0, t^* \in [0, \bar{T}_{\epsilon^*}], s^* \in \mathcal{S}_{t^*}^{s_0, \Pi^{MD}}, \boldsymbol{\pi}_* \in \Pi^{MD}$ s.t.

$$\mathbb{P}[S_{t^*}^{\boldsymbol{\pi}_*} = s^* | s_0] > 0 \tag{25}$$

and $\forall \kappa > 0$,

$$\frac{\epsilon^*}{\kappa} < \|V^{\boldsymbol{\pi}_*}(s^*) - V^{\boldsymbol{\pi}_*^{\bar{T}_{\epsilon^*} - t^*}} \cdot \ddot{\boldsymbol{\pi}}(s_{t^*})\| = \sum_{\tau = \bar{T}_{\epsilon^*} - t^* + 1}^{\infty} \mathbb{P}[T_{\bar{\mathcal{S}}}^{\boldsymbol{\pi}_*} = \tau | s^*] \mathbb{E}[\varphi(\tau) R_\tau^{\boldsymbol{\pi}_*} | s^*] \tag{26}$$

Let us fix $\boldsymbol{\pi} \coloneqq \boldsymbol{\pi}_*$ and set

$$\kappa^* \coloneqq \mathbb{P}[S_{t^*}^{\boldsymbol{\pi}_*} = s^* | s_0] \varphi(\bar{T}_{\epsilon^*} + 1) \tag{27}$$

Note that $\kappa^* > 0$ by (25) and $\bar{T}_{\epsilon^*} < \infty$ (by Assumption 3.3, $\bar{T}_{s_0, \epsilon^*} < \infty$, and by Assumption 3.4, $\hat{T}_{s_0} < \infty$). Then,

$$\left| V^{\boldsymbol{\pi}_*}(s_0) - V^{\boldsymbol{\pi}_*^{\bar{T}_{s_0, \epsilon^*}}} \cdot \ddot{\boldsymbol{\pi}}(s_0) \right| = \sum_{\tau = \bar{T}_{s_0, \epsilon^*} + 1} \mathbb{P}[T_{\bar{\mathcal{S}}}^{\boldsymbol{\pi}_*} = \tau | s_0] \mathbb{E}[\varphi(\tau) R_\tau^{\boldsymbol{\pi}_*} | s_0]$$

$$\text{(by boundary-only rewards; see Lemma A.2's proof)}$$

$$\geq \sum_{\tau = \bar{T}_{\epsilon^*} + 1} \mathbb{P}[T_{\bar{\mathcal{S}}}^{\boldsymbol{\pi}_*} = \tau | s_0] \mathbb{E}[\varphi(\tau) R_\tau^{\boldsymbol{\pi}_*} | s_0] \qquad (\text{by } \bar{T}_{s_0, \epsilon^*} \leq \bar{T}_{\epsilon^*})$$

$$= \sum_{\tau = \bar{T}_{\epsilon^*} + 1} \sum_{s \in \mathcal{S}} \mathbb{P}[S_{t^*}^{\boldsymbol{\pi}_*} = s | s_0] \mathbb{P}[T_{\bar{\mathcal{S}}}^{\boldsymbol{\pi}_*} = \tau - t^* | s] \mathbb{E}[\varphi(\tau) R_{\tau - t^*}^{\boldsymbol{\pi}_*} | s]$$

$$\geq \mathbb{P}[S_{t^*}^{\boldsymbol{\pi}_*} = s^* | s_0] \sum_{\tau = \bar{T}_{\epsilon^*} - t^* + 1} \mathbb{P}[T_{\bar{\mathcal{S}}}^{\boldsymbol{\pi}_*} = \tau | s^*] \mathbb{E}[\varphi(\tau + t^*) R_\tau^{\boldsymbol{\pi}_*} | s^*]$$

$$\text{(by non-negative rewards and probabilities)}$$

$$\geq \mathbb{P}[S_{t^*}^{\boldsymbol{\pi}_*} = s^* | s_0] \varphi(\bar{T}_{\epsilon^*} + 1) \sum_{\tau = \bar{T}_{\epsilon^*} - t^* + 1} \mathbb{P}[T_{\bar{\mathcal{S}}}^{\boldsymbol{\pi}_*} = \tau | s^*] \mathbb{E}[\varphi(\tau) R_\tau^{\boldsymbol{\pi}_*} | s^*]$$

$$(\text{by (24) and } t^* \in [0, \bar{T}_{\epsilon^*}])$$

$$> \mathbb{P}[S_{t^*}^{\boldsymbol{\pi}_*} = s^* | s_0] \varphi(\bar{T}_{\epsilon^*} + 1) \frac{\epsilon^*}{\kappa^*} = \epsilon^* \qquad (\text{by (26) and (27)})$$

This contradicts definition of $\bar{T}_{s_0, \epsilon^*}$ (see Assumption 3.3), implying that our supposition is false.

With $\kappa(\epsilon, t, s_t, \boldsymbol{\pi}; s_0) \coloneqq \mathbb{P}[S_t^{\boldsymbol{\pi}} = s_t | s_0] \varphi(\bar{T}_\epsilon + 1)$, we will now show that

$$\lim_{\epsilon \to 0} \frac{\epsilon}{\mathbb{P}[S_t^{\boldsymbol{\pi}} = s_t | s_0] \varphi(\bar{T}_\epsilon + 1)} = 0 \tag{28}$$

For any fixed $\epsilon > 0$, let us define

$$G(\bar{T}_\epsilon; s_0) \coloneqq \min\{\mathbb{P}[S_t^{\boldsymbol{\pi}} = s_t | s_0] > 0 : t \in [0, \bar{T}_\epsilon], s_t \in \mathcal{S}, \boldsymbol{\pi} \in \Pi^{MD}\}. \tag{29}$$

Then, $\forall \boldsymbol{\pi} \in \Pi^{MD}, \forall t \in [0, \bar{T}_\epsilon], \forall s_t \in \mathcal{S}_t^{s_0, \Pi^{MD}}$,

$$0 \leq \frac{\epsilon}{\mathbb{P}[S_t^{\boldsymbol{\pi}} = s_t | s_0] \varphi(\bar{T}_\epsilon + 1)} \leq \frac{\epsilon}{G(\bar{T}_\epsilon; s_0) \varphi(\bar{T}_\epsilon + 1)} \tag{30}$$

Since $\bar{T}_{s_0, 0} < \infty$, we have

$$\bar{T}_0 \doteq \max\{\hat{T}_{s_0}, \bar{T}_{s_0, 0}\} < \infty. \tag{31}$$

Let us fix arbitrarily $\boldsymbol{\pi} \in \Pi^{MD}, t \in [0, \bar{T}_0], s_t \in \mathcal{S}_t^{s_0, \Pi^{MD}}$. By (31), $\lim_{\epsilon \to 0} G(\bar{T}_\epsilon; s_0) = G(\bar{T}_0; s_0) > 0$ and $\lim_{\epsilon \to 0} \varphi(\bar{T}_\epsilon + 1) = \varphi(\bar{T}_0 + 1) > 0$. Thus, we can take $\lim_{\epsilon \to 0}$ on the upper and lower bound in (30) and have shown

$$\lim_{\epsilon \to 0} \frac{\epsilon}{\mathbb{P}[S_t^{\boldsymbol{\pi}} = s_t | s_0] \varphi(\bar{T}_\epsilon + 1)} = 0$$

$\square$

Finally, it's straightforward to verify that our hyperbolic Gridworld in Figure 1(a) has *boundary-only* rewards and $s_0 = 21$ that satisfies Assumption 3.4. Moreover, due to the existence of $\tau^* := \max\{T_{\bar{\mathcal{S}}}^{\boldsymbol{\pi}} < \infty : \boldsymbol{\pi} \in \Pi^{MD}\} < \infty$ by its deterministic transition and $|\Pi^{MD}| < \infty$, we have $\forall \bar{T} \geq \tau^*$,

$$\sup_{\boldsymbol{\pi} \in \Pi^{MD}} |V^{\boldsymbol{\pi}}(s_0) - V^{\boldsymbol{\pi}^{\bar{T}} \cdot \tilde{\boldsymbol{\pi}}}(s_0)| = \sup_{\boldsymbol{\pi} \in \Pi^{MD}: T_{\bar{\mathcal{S}}}^{\boldsymbol{\pi}} < \infty} |V^{\boldsymbol{\pi}}(s_0) - V^{\boldsymbol{\pi}^{\bar{T}} \cdot \tilde{\boldsymbol{\pi}}}(s_0)| \qquad \text{(by boundary-only rewards)}$$

$$= 0 \qquad (\text{ by } \forall \tau > \bar{T} \geq \tau^*, \forall \boldsymbol{\pi} \in \Pi^{MD} \text{ with } \bar{T}_{\bar{\mathcal{S}}}^{\boldsymbol{\pi}} < \infty, \mathbb{P}[T_{\bar{\mathcal{S}}}^{\boldsymbol{\pi}} = \tau | s_0] = 0)$$

and thus, $\bar{T}_{s_0, 0} \leq \tau^* < \infty$. For a more concrete example, we can refer to the *restricted* Hyperbolic Gridworld in Example 4.6, where we can compute manually $\hat{T}_{s_0} = 7$ and $\bar{T}_{s_0, 0} = 8$.

## A.2 Implied Bounded Value Functions

Through the following lemma, we can link Assumption 3.3 to the standard well-posedness condition of bounded value functions that ensures the existence of optimal policy.

**Lemma A.4.** *If Assumption 3.3 holds, then $\forall s \in \mathcal{S}, \forall \boldsymbol{\pi} \in \Pi^{MD}, V^{\boldsymbol{\pi}}(s) < \infty$.*

*Proof.* Suppose $\exists \boldsymbol{\pi}_*, s_*$ s.t. $V^{\boldsymbol{\pi}_*}(s_*) = \infty$. Then, we can set $s, \boldsymbol{\pi} \leftarrow s_*, \boldsymbol{\pi}_*$ and arbitrary $\epsilon^* > 0$ s.t. $\forall \bar{T} < \infty$, $|V^{\boldsymbol{\pi}_*^{\bar{T}} \cdot \tilde{\boldsymbol{\pi}}}(s_*) - V^{\boldsymbol{\pi}_*}(s_*)| > \epsilon^*$ since $V^{\boldsymbol{\pi}_*^{\bar{T}} \cdot \tilde{\boldsymbol{\pi}}}(s_*) < \infty$. $\square$

# B Theorem 4.3

## B.1 Technical Assumptions

**Assumption B.1** ("Relevant at $t$ under $\boldsymbol{\pi}$"). If $t = 0$,

$$\mathbb{P}[S_t^{\boldsymbol{\pi}} = s_0 | s_0] = 0, \forall t \geq 1 \wedge \mathbb{P}[S_1 = s_0 | S_0 = s_0, A_0 = a_0] = 0 \tag{32}$$

If $t > 0$, $\exists (s_{t-1}, a_{t-1})$ "relevant at $t - 1$ under $\boldsymbol{\pi}$" s.t.

$$\mathbb{P}[S_t = s_t | S_{t-1} = s_{t-1}, A_{t-1} = a_{t-1}] > 0 \wedge a_t = \pi(s_t). \tag{33}$$

Intuitively, for $t > 0$, (33) exhausts the use instances of $(s_t, a_t)$ in PE updates $Q^{\boldsymbol{\pi}}(s_{t-1}, a_{t-1}) \leftarrow \mathbb{E}[Q^{\boldsymbol{\pi}}(s_t, \pi(s_t))] + \dots$ and thus, it must hold. Whereas for $t = 0$, some restrictions on the MDP (e.g., $\forall t' \neq t, \mathcal{S}_t^{s_0, \Pi^{MD}} \cap \mathcal{S}_{t'}^{s_0, \Pi^{MD}} = \emptyset$ as in Example 4.6) can be imposed to ensure that (32) holds $\forall \boldsymbol{\pi} \in \Pi^{MD}, s_0 \in \mathcal{S} \setminus \{\bar{\mathcal{S}}\}, a_0 \in \mathcal{A}_{s_0}$. Note however that in actual use instances, (32) only need to hold for the $\boldsymbol{\pi}$ encountered in the PI updates (instead of $\forall \boldsymbol{\pi} \in \Pi^{MD}$). This may relax the need for such MDP restrictions: as we can observe from our experiments (see Section 6), our algorithm still performs plausibly well even when $\forall t' \neq t, \mathcal{S}_t^{s_0, \Pi^{MD}} \cap \mathcal{S}_{t'}^{s_0, \Pi^{MD}} = \emptyset$ does not hold. In what follows, we present several intermediate results that link the conditions in Assumption B.1 to the "approximation" (13) in Theorem 4.3.

**Lemma B.2.** *At any $t \geq 0$, if $(s_t, a_t)$ is "relevant at $t$ under $\boldsymbol{\pi}$", then $\exists s_0, a_0$ "relevant at 0 under $\boldsymbol{\pi}$" s.t.*

$$\mathbb{P}[S_t^{\boldsymbol{\pi}_{s_0, a_0}} = s_t | s_0] > 0 \wedge a_t = \pi_{s_0, a_0}(s_t)$$

*with $\boldsymbol{\pi}_{s_0,a_0}$ defined as follows*

$$\pi_{s_0,a_0}(s) = \begin{cases} a_0, & if\ s = s_0 \\ \pi(s), & otherwise \end{cases} \tag{34}$$

*Proof.* (Base case: $t = 0$.) Note that for any $(s_0, a_0)$ that is "relevant at 0 under $\boldsymbol{\pi}$", we have $\mathbb{P}[S_0^{\boldsymbol{\pi}_{s_0,a_0}}|s_0] = 1 > 0$. Moreover, $a_0 = \pi_{s_0,a_0}(s_0)$ holds by definition in (34).

($t > 0$.) Proof by induction. Suppose that the relation holds at $t = t' - 1$, we will show that it also holds at $t = t'$. By $(s_{t'}, a_{t'})$'s "relevance at $t'$ under $\boldsymbol{\pi}$", $\exists (s_{t'-1}, a_{t'-1})$ "relevant at $t' - 1$ under $\boldsymbol{\pi}$" s.t.

$$\mathbb{P}[S_{t'} = s_{t'}|S_{t'-1} = s_{t'-1}, A_{t'-1} = a_{t'-1}] > 0 \wedge a_{t'} = \pi(s_{t'}) \tag{35}$$

Moreover, by assumption (that at $t = t' - 1$ the relation holds), the above $(s_{t'-1}, a_{t'-1})$ satisfies: $\exists s_0, a_0$ "relevant at 0 under $\boldsymbol{\pi}$ s.t.

$$\mathbb{P}[S_{t'-1}^{\boldsymbol{\pi}_{s_0,a_0}} = s_{t'-1}|s_0] > 0 \wedge a_{t'-1} = \pi_{s_0,a_0}(s_{t'-1}). \tag{36}$$

Therefore,

$$\begin{aligned}
\mathbb{P}[S_{t'}^{\boldsymbol{\pi}_{s_0,a0}} = s_{t'}|s_0] &\geq \mathbb{P}[S_{t'}^{\boldsymbol{\pi}_{s_0,a0}} = s_{t'}|S_{t'-1} = s_{t'-1}]\mathbb{P}[S_{t'-1}^{\boldsymbol{\pi}_{s_0,a0}} = s_{t'-1}|s_0] \\
&= \mathbb{P}[S_{t'} = s_{t'}|S_{t'-1} = s_{t'-1}, A_{t'-1} = a_{t'-1}]\mathbb{P}[S_{t'-1}^{\boldsymbol{\pi}_{s_0,a0}} = s_{t'-1}|s_0] > 0 \qquad \text{(by (36))}
\end{aligned}$$

Moreover, by $(s_0, a_0)$'s "relevance at 0 under $\boldsymbol{\pi}$" and $t' > 0$, we must have $s_{t'} \neq s_0$ which then implies

$$a_{t'} = \pi_{s_0,a_0}(s_{t'}) \qquad \text{(by (35))}$$

$\square$

**Lemma B.3.** *For any $\boldsymbol{\pi} \in \Pi^{MD}$, $t \geq 0$, and $(s_t, a_t)$ "relevant at $t$ under $\boldsymbol{\pi}$", $\exists \kappa > 0$ s.t.*

$$\forall s_{t+1} \sim p_{s_t}^{a_t}, \left| V^{\boldsymbol{\pi}}(s_{t+1}) - V^{\boldsymbol{\pi}^{\bar{T}_\epsilon - (t+1)} \cdot \ddot{\boldsymbol{\pi}}}(s_{t+1}) \right| \leq \frac{\epsilon}{\kappa} \tag{37}$$

*and $\lim_{\epsilon \downarrow 0} \frac{\epsilon}{\kappa} = 0$.*

*Proof.* Let us first fix arbitrarily $(s_t, a_t, \boldsymbol{\pi})$. By Lemma B.2, $\exists s_0, a_0$ and $\tilde{\boldsymbol{\pi}} := \boldsymbol{\pi}_{s_0,a_0}$ s.t.

$$\mathbb{P}[S_t^{\tilde{\boldsymbol{\pi}}} = s_t|s_0] > 0 \wedge a_t = \tilde{\pi}(s_t) \tag{38}$$

Next, for any arbitrary choice of $s_{t+1} \sim p_{s_t}^{a_t}$, we have

$$\mathbb{P}[S_{t+1} = s_{t+1}|S_t = s_t, A_t = a_t] > 0 \tag{39}$$

Therefore,

$$\begin{aligned}
\mathbb{P}[S_{t+1}^{\tilde{\boldsymbol{\pi}}} = s_{t+1}|s_0] &\geq \mathbb{P}[S_{t+1}^{\tilde{\boldsymbol{\pi}}} = s_{t+1}|S_t = s_t]\mathbb{P}[S_t^{\tilde{\boldsymbol{\pi}}} = s_t|s_0] \\
&= \mathbb{P}[S_{t+1} = s_{t+1}|S_t = s_t, A_t = a_t]\mathbb{P}[S_t^{\tilde{\boldsymbol{\pi}}} = s_t|s_0] \qquad \text{(by (38))} \\
&> 0 \qquad \text{(by (38) and (39))}
\end{aligned}$$

which by Assumption 3.5, implies

$$\begin{aligned}
\frac{\epsilon}{\kappa(\epsilon, t+1, s_{t+1}, \tilde{\boldsymbol{\pi}}; s_0)} &\geq \left| V^{\tilde{\boldsymbol{\pi}}}(s_{t+1}) - V^{\tilde{\boldsymbol{\pi}}^{\bar{T}_\epsilon - (t+1)} \cdot \ddot{\boldsymbol{\pi}}}(s_{t+1}) \right| \\
&= \left| V^{\boldsymbol{\pi}}(s_{t+1}) - V^{\boldsymbol{\pi}^{\bar{T} - (t+1)} \cdot \ddot{\boldsymbol{\pi}}}(s_{t+1}) \right|
\end{aligned}$$

(by $(s_0, a_0)$'s "relevance at 0 under $\boldsymbol{\pi}$" and $t + 1 > 0$, $s_{t+1} \neq s_0$.)

and

$$\lim_{\epsilon \downarrow 0} \frac{\epsilon}{\kappa(\epsilon, t+1, s_{t+1}, \tilde{\boldsymbol{\pi}}; s_0)} = 0. \tag{40}$$

Now, let us choose $\kappa \coloneqq \min\{\kappa(\epsilon, t+1, s_{t+1}, \tilde{\boldsymbol{\pi}}; s_0) : s_{t+1} \sim p_{s_t}^{a_t}\}$. Note that (37) directly holds. It thus remains to show the following,

$$\begin{aligned}
\lim_{\epsilon \downarrow 0} \frac{\epsilon}{\kappa} &= \lim_{\epsilon \downarrow 0} \frac{\epsilon}{\min\{\kappa(\epsilon, t+1, s_{t+1}, \tilde{\boldsymbol{\pi}}; s_0) : s_{t+1} \sim p_{s_t}^{a_t}\}} \\
&= \lim_{\epsilon \downarrow 0} \max \left\{ \frac{\epsilon}{\kappa(\epsilon, t+1, s_{t+1}, \tilde{\boldsymbol{\pi}}; s_0)} : s_{t+1} \sim p_{s_t}^{a_t} \right\} \\
&= 0 \qquad \qquad \text{(by (40) and at most finitely many choices of } s_{t+1} \in \mathcal{S})
\end{aligned}$$

$\square$

## B.2   Proof of Theorem 4.3

For any $m \geq \tau + 1$, we can derive r-function recursion as follows

$$r^{\boldsymbol{\pi}, \tau, m}(s, a) \doteq \mathbb{E}\left[\varphi(m-\tau)R\left(S_m^{a \cdot \boldsymbol{\pi}}, \pi(S_m^{a \cdot \boldsymbol{\pi}})\right) | S_\tau = s\right] \tag{41}$$

$$= \mathbb{E}_{S_{\tau+1} \sim p_s^a}\left[\mathbb{E}\left[\varphi(m-\tau)R\left(S_m^{\boldsymbol{\pi}}, \pi(S_m^{\boldsymbol{\pi}})\right) | S_{\tau+1}\right]\right] \tag{42}$$

$$= \mathbb{E}_{S_{\tau+1} \sim p_s^a}\left[\frac{\varphi(m-\tau)}{\varphi(m-(\tau+1))}\mathbb{E}\left[\varphi(m-(\tau+1))R\left(S_m^{\boldsymbol{\pi}}, \pi(S_m^{\boldsymbol{\pi}})\right) | S_{\tau+1}\right]\right] \tag{43}$$

$$= \mathbb{E}_{S_{\tau+1} \sim p_s^a}\left[\frac{\varphi(m-\tau)}{\varphi(m-(\tau+1))}r^{\boldsymbol{\pi}, \tau+1, m}(S_{\tau+1}, \pi(S_{\tau+1}))\right] \tag{44}$$

For $m = \tau$, by Definition 4.2, we have

$$r^{\boldsymbol{\pi}, \tau, \tau}(s, a) \doteq \mathbb{E}\left[\varphi(m-\tau)R\left(S_m^{a \cdot \boldsymbol{\pi}}, \boldsymbol{\pi}(S_m^{a \cdot \boldsymbol{\pi}})\right) | S_\tau = s\right] \tag{45}$$

$$= \mathbb{E}_{R \sim p_s^a}[\varphi(0)R(s, a)] \tag{46}$$

Next, we derive Q-function recursion,

$$Q^{\boldsymbol{\pi}}(s_t, a_t) \doteq \mathbb{E}\left[\varphi(0)R(s_t, a_t) + \varphi(1)R(S_{t+1}^{a_t \cdot \boldsymbol{\pi}}, \boldsymbol{\pi}(S_{t+1}^{a_t \cdot \boldsymbol{\pi}})) + \ldots | S_t = s_t\right] \tag{47a}$$

$$\begin{aligned}
&= \mathbb{E}_{R \sim p_{s_t}^{a_t}}[\varphi(0)R(s_t, a_t)] + \mathbb{E}\left[\varphi(0)R(S_{t+1}^{a_t \cdot \boldsymbol{\pi}}, \boldsymbol{\pi}(S_{t+1}^{a_t \cdot \boldsymbol{\pi}})) + \varphi(1)R(S_{t+2}^{a_t \cdot \boldsymbol{\pi}}, \boldsymbol{\pi}(S_{t+2}^{a_t \cdot \boldsymbol{\pi}})) + \ldots | S_t = s_t\right] \\
&\quad - \left\{\mathbb{E}\left[\varphi(0)R(S_{t+1}^{a_t \cdot \boldsymbol{\pi}}, \boldsymbol{\pi}(S_{t+1}^{a_t \cdot \boldsymbol{\pi}})) + \varphi(1)R(S_{t+2}^{a_t \cdot \boldsymbol{\pi}}, \boldsymbol{\pi}(S_{t+2}^{a_t \cdot \boldsymbol{\pi}})) + \ldots | S_t = s_t\right] \right. \\
&\quad \left. - \mathbb{E}\left[\varphi(1)R(S_{t+1}^{a_t \cdot \boldsymbol{\pi}}, \boldsymbol{\pi}(S_{t+1}^{a_t \cdot \boldsymbol{\pi}})) + \varphi(2)R(S_{t+2}^{a_t \cdot \boldsymbol{\pi}}, \boldsymbol{\pi}(S_{t+2}^{a_t \cdot \boldsymbol{\pi}})) + \ldots | S_t = s_t\right]\right\}
\end{aligned} \tag{47b}$$

$$\begin{aligned}
&= \mathbb{E}_{R \sim p_{s_t}^{a_t}}[\varphi(0)R(s_t, a_t)] + \mathbb{E}_{S_{t+1} \sim p_{s_t}^{a_t}}[Q^{\boldsymbol{\pi}}(S_{t+1}, \boldsymbol{\pi}(S_{t+1}))] \\
&\quad - \left\{\mathbb{E}_{S_{t+1} \sim p_{s_t}^{a_t}}\left[\mathbb{E}\left[\varphi(0)R(S_{t+1}^{\boldsymbol{\pi}}, \boldsymbol{\pi}(S_{t+1}^{\boldsymbol{\pi}})) + \varphi(1)R(S_{t+2}^{\boldsymbol{\pi}}, \boldsymbol{\pi}(S_{t+2}^{\boldsymbol{\pi}})) + \ldots | S_{t+1}\right]\right] \right. \\
&\quad \left. - \mathbb{E}_{S_{t+1} \sim p_{s_t}^{a_t}}\left[\mathbb{E}\left[\varphi(1)R(S_{t+1}^{\boldsymbol{\pi}}, \boldsymbol{\pi}(S_{t+1}^{\boldsymbol{\pi}})) + \varphi(2)R(S_{t+2}^{\boldsymbol{\pi}}, \boldsymbol{\pi}(S_{t+2}^{\boldsymbol{\pi}})) + \ldots | S_{t+1}\right]\right]\right\}
\end{aligned} \tag{47c}$$

$$\begin{aligned}
&= \mathbb{E}_{R \sim p_{s_t}^{a_t}}[\varphi(0)R(s_t, a_t)] + \mathbb{E}_{S_{t+1} \sim p_{s_t}^{a_t}}[Q^{\boldsymbol{\pi}}(S_{t+1}, \boldsymbol{\pi}(S_{t+1}))] \\
&\quad - \left\{\mathbb{E}_{S_{t+1} \sim p_{s_t}^{a_t}}\left[\mathbb{E}\left[\sum_{m=t+1}^{\infty} \varphi(m-(t+1))R(S_m^{\boldsymbol{\pi}}, \boldsymbol{\pi}(S_m^{\boldsymbol{\pi}})) \Big| S_{t+1}\right]\right] \right. \\
&\quad \left. - \mathbb{E}_{S_{t+1} \sim p_{s_t}^{a_t}}\left[\mathbb{E}\left[\sum_{m=t+1}^{\infty} \varphi(m-t)R(S_m^{\boldsymbol{\pi}}, \boldsymbol{\pi}(S_m^{\boldsymbol{\pi}})) \Big| S_{t+1}\right]\right]\right\}
\end{aligned} \tag{47d}$$

On the 2nd line, we apply $\forall s_{t+1} \sim p_{s_t}^{a_t}$,

$$\left| \mathbb{E}\left[ \sum_{m=t+1}^{\infty} \varphi(m-(t+1))R_m^{\pi}|S_{t+1}=s_{t+1}\right] - \mathbb{E}\left[ \sum_{m=t+1}^{\bar{T}_\epsilon} \varphi(m-(t+1))R_m^{\pi}|S_{t+1}=s_{t+1}\right] \right|$$

$$= \left| \mathbb{E}\left[ \sum_{m=0}^{\infty} \varphi(m)R_m^{\pi}|S_0=s_{t+1}\right] - \mathbb{E}\left[ \sum_{m=0}^{\bar{T}_\epsilon-(t+1)} \varphi(m)R_m^{\pi}|S_0=s_{t+1}\right] \right|$$

$$= |V^{\pi}(s_{t+1}) - V^{\pi^{\bar{T}_\epsilon-(t+1)}\cdot\ddot{\pi}}(s_{t+1})|$$

$$\leq \frac{\epsilon}{\kappa} \qquad\qquad\qquad\qquad\qquad\qquad\qquad \text{(by using } \kappa \text{ from Lemma B.3)}$$

On the 3rd line, we apply $\forall s_{t+1} \sim p_{s_t}^{a_t}$,

$$\left| \mathbb{E}\left[ \sum_{m=t+1}^{\infty} \varphi(m-t)R_m^{\pi}||S_{t+1}=s_{t+1}\right] - \mathbb{E}\left[ \sum_{m=t+1}^{\bar{T}_\epsilon} \varphi(m-t)R_m^{\pi}|S_{t+1}=s_{t+1}\right] \right|$$

$$= \left| \mathbb{E}\left[ \sum_{m=1}^{\infty} \varphi(m)R_{m-1}^{\pi}|S_0=s_{t+1}\right] - \mathbb{E}\left[ \sum_{m=1}^{\bar{T}_\epsilon-t} \varphi(m)R_{m-1}^{\pi}|S_0=s_{t+1}\right] \right|$$

$$\leq \left| \mathbb{E}\left[ \sum_{m=1}^{\infty} \varphi(m-1)R_{m-1}^{\pi}|S_0=s_{t+1}\right] - \mathbb{E}\left[ \sum_{m=1}^{\bar{T}_\epsilon-t} \varphi(m-1)R_{m-1}^{\pi}|S_0=s_{t+1}\right] \right| \quad \text{(by } \varphi(.) \text{ decreasing)}$$

$$= \left| V^{\pi}(s_{t+1}) - V^{\pi^{\bar{T}_\epsilon-(t+1)}\cdot\ddot{\pi}}(s_{t+1}) \right| \qquad\qquad\qquad\qquad\qquad\qquad \text{(by (5))}$$

$$\leq \frac{\epsilon}{\kappa} \qquad\qquad\qquad\qquad\qquad\qquad\qquad\qquad\qquad \text{(by using } \kappa \text{ from Lemma B.3)}$$

Therefore, continuing from (47d), we can perform approximation with $\bar{T}_\epsilon < \infty$ as follows,

$$\approx \mathbb{E}_{R \sim p_{s_t}^{a_t}}[\varphi(0)R(s_t, a_t)] + \mathbb{E}_{S_{t+1}\sim p_{s_t}^{a_t}}\left[Q^{\pi}(S_{t+1}, \pi(S_{t+1}))\right]$$

$$- \left\{ \mathbb{E}_{S_{t+1}\sim p_{s_t}^{a_t}}\left[ \sum_{m=t+1}^{\bar{T}_\epsilon} \mathbb{E}\left[\varphi(m-(t+1))R(S_m^{\pi}, \pi(S_m^{\pi}))|S_{t+1}\right]\right] \right.$$

$$\left. - \mathbb{E}_{S_{t+1}\sim p_{s_t}^{a_t}}\left[ \sum_{m=t+1}^{\bar{T}_\epsilon} \frac{\varphi(m-t)}{\varphi(m-(t+1))}\mathbb{E}\left[\varphi(m-(t+1))R(S_m^{\pi}, \pi(S_m^{\pi}))|S_{t+1}\right]\right] \right\}$$

$$\text{(by setting } \kappa(\epsilon) = \kappa/2 \text{ which directly implies } \lim_{\epsilon\downarrow 0} \frac{\epsilon}{\kappa(\epsilon)} = 0\text{)}$$

By applying (46), Definition 4.2, and (44) on the 1st, 2nd, and 3rd line, respectively, we can then obtain

$$= \mathbb{E}_{R\sim p_{s_t}^{a_t}}[\varphi(0)R(s_t, a_t)] + \mathbb{E}_{S_{t+1}\sim p_{s_t}^{a_t}}\left[Q^{\pi}(S_{t+1}, \pi(S_{t+1}))\right]$$

$$- \left\{ \sum_{m=t+1}^{\bar{T}_\epsilon} \left( \mathbb{E}_{S_{t+1}\sim p_{s_t}^{a_t}}\left[r^{\pi,t+1,m}(S_{t+1}, \pi(S_{t+1}))\right] - r^{\pi,t,m}(s_t, a_t)\right) \right\} \qquad (47e)$$

Finally, based on the Definition 3.6, we will set our boundary conditions (when we are at some *boundary* states),

$$Q^{\pi}(s_t, a_t) = \mathbb{E}_{R\sim p_{s_t}^{a_t}}[\varphi(0)R(s_t, a_t)], \forall s_t \in \bar{\mathcal{S}}, a_t \in \bar{\mathcal{A}} \qquad (48)$$

## C  Backward Q-learning Algorithm

In this section, we detail the derivations of our Backward Q-learning in Section 5.2 from Theorem 4.3.

### C.1 r-table Update

Based on the $r$-function recursion, i.e. (44) and (46), we obtain bootstrap targets that corresponds to (20) in the main paper,

$$\xi_t^r(m) \leftarrow \varphi(0)R(s_t, a_t), \quad \text{for } m = t \tag{49}$$

$$\xi_t^r(m) \leftarrow \frac{\varphi(m-t)}{\varphi(m-(t+1))} r^{t+1,m}(S_{t+1}, \pi'(S_{t+1})), \quad \text{for } m = t+1 : T^* \tag{50}$$

Then, updates to $r$-table are made as follows,

$$r^{t,m}(s_t, a_t) \leftarrow (1 - \alpha_r)r^{t,m}(s_t, a_t) + \alpha_r \xi_t^r(m), \quad \text{for } m = t : T^* \tag{51}$$

given learning rate $\alpha_r > 0$.

### C.2 Q-table Update

Based on the Q-function recursion, i.e. (47e) and (48), we obtain bootstrap targets that corresponds to (21) in the main paper,

$$\xi_t^Q \leftarrow \gamma(0)R(s_t, a_t), \quad \text{for } t = T^* \text{ and } s_t \in \bar{\mathcal{S}} \tag{52}$$

$$\xi_t^Q \leftarrow \gamma(0)R(s_t, a_t) + Q(S_{t+1}, \pi'(S_{t+1})) - \max(0, \Delta r_t), \quad \text{for } t \leq T^* - 1 \tag{53}$$

where $\Delta r_t = \sum_{m=t+1}^{T^*} r^{t+1,m}(s_{t+1}, \pi'(s_{t+1})) - r^{t,m}(s_t, a_t)$. Then, updates to $Q$-table can be done as follows,

$$Q(s_t, a_t) \leftarrow (1 - \alpha_Q)Q(s_t, a_t) + \alpha_Q \xi_t^Q, \quad \text{for } t \leq T^* \tag{54}$$

given learning rate $\alpha_Q > 0$.

**Truncation from $\bar{T}$ to $T^*$.** For our implementation, instead of keeping track of all values up to $\bar{T}$, we use the variable length $T^*$ of each trajectory sampled following a current policy $\boldsymbol{\pi}$. However, we will still set a sufficiently large $\bar{T}$ as a proxy for $\bar{T}_\epsilon$ to ensure that all trajectory terminates.

**Clipping of adjustment terms.** Let us denote by $\Delta r_t^{\boldsymbol{\pi}}$ the adjustment terms in the 2nd row of (47e) as in the main paper. Referring to (53), we note that the clipped function $\max(0, \Delta r_t)$ has been used in place of $\Delta r_t$. This is done to slow down the accumulation of error relevant to $\Delta r_t \approx \Delta r_t^{\boldsymbol{\pi}}$. In particular, we note that $\Delta r_t^{\boldsymbol{\pi}} \geq 0$:

$$\Delta r_t^{\boldsymbol{\pi}} \doteq \sum_{m=t+1}^{\bar{T}-1} \left( \mathbb{E}_{S' \sim p_{s_t}^{a_t}} \left[ r^{\boldsymbol{\pi}, t+1, m}(S', \pi(S')) \right] - r^{\boldsymbol{\pi}, t, m}(s_t, a_t) \right)$$

$$= \sum_{m=t+1}^{\bar{T}-1} \left( \mathbb{E}_{S' \sim p_{s_t}^{a_t}} \left[ \left( 1 - \frac{\varphi(m-t)}{\varphi(m-(t+1))} \right) r^{\boldsymbol{\pi}, t+1, m}(S', \pi(S')) \right] \right) \quad \text{(by (44))}$$

$$\geq 0 \qquad \text{(by } \varphi(\cdot) \text{ discount factor and } R : \mathcal{S} \times \mathcal{A} \to \mathbb{R}^+ \text{ s.t. (9) is non-negative)}$$

But without clipping, $\Delta r_t < 0$ may happen in subsequent iterations, inflating Q-values at some states past a certain threshold such that their neighboring states prefer transition to these inflated states than moving towards goal states, when the latter clearly results in a fewer steps. This then creates a looping behaviour which eventually lead to divergence.

### C.3 Policy-table Update

We note our use of policy-table separate from the $\arg\max$ of a Q-table to represent *greedy* policy. This is due to the possibilities of *non-unique* actions realizing $\arg\max_a Q(s, a)$ for some $s \in \mathcal{S}$ which may cause non-unique r-function related values, i.e. the components in $\sum_{m=t+1}^{T^*} \Delta r_t^m$, after substituting *different* global optima actions. Specifically, we follow the *consistent tie-break* rule proposed in Section 3.3 of Lesmana & Pun (2021); see line 18-22 in Algorithm 1.

---

**Algorithm 2** On-policy Monte Carlo Control (MC)

---

1: **Input:** Hyperbolic ($k = 1$) Gridworld, Hyperparameters ($\epsilon, \bar{T}$)
2: **Output:** Approximate SPE Q-function $Q^{\hat{\pi}}(s, a) \forall s \in \mathcal{S}, a \in \mathcal{A}$
3: **Initialize:** $Q(s, a) \leftarrow 0, \forall s \in \mathcal{S} \setminus \bar{\mathcal{S}}, a \in \mathcal{A}$; $Q(s, a) \leftarrow \varphi(0) R(s, a), \forall s \in \bar{\mathcal{S}}, a \in \bar{\mathcal{A}}$; $Returns(s, a) \leftarrow \emptyset, \forall s \in \mathcal{S}, a \in \mathcal{A}(s)$; $\pi'(s) \leftarrow \arg\max_a Q(s, a), \forall s \in \mathcal{S}$; $\boldsymbol{\pi} \leftarrow \emptyset$;
4: **repeat**
5:     Update $\boldsymbol{\pi} \leftarrow \boldsymbol{\pi}'$
6:     Choose $S_0$ randomly
7:     Generate *trajectory* $\omega_{0:T^*} \doteq S_0, A_0, \dots, S_{T^*}, A_{T^*}$ following $\boldsymbol{\pi}^\epsilon$
8:     Set $G \leftarrow 0$
9:     **for** $t \leftarrow 0$ **to** $T^*$ **do**
10:       **if** the pair $(S_t, A_t)$ does not appear in $\omega_{0:t-1}$ **then**
11:         Compute $G \leftarrow \varphi(T^* - t) R(S_{T^*}, A_{T^*})$
12:         Append $G$ to $Returns(S_t, A_t)$
13:         Update $Q(S_t, A_t) \leftarrow \text{average}(Returns(S_t, A_t))$
14:         Update $\pi'(S_t) \leftarrow \arg\max_a Q(S_t, a)$
15:       **end if**
16:     **end for**
17: **until** stable ($\boldsymbol{\pi}' \neq \boldsymbol{\pi}$)

---

# D  NUMERICAL EXAMPLES

This section provides some missing details on Section 6.

## D.1  Environment Setup

We review 3 important considerations in our Gridworld designs: (i) existence of actual *future deviations* (i.e. if states like $(s_0, s_\tau) = (21, 9)$ exist, where the optimality of 21's action is constrained by 9's action such that we have *priority ordering* on $\mathcal{S}$), (ii) $\pi^{*0}(s_0) \neq \hat{\pi}(s_0)$ where the value of following SPE path $\hat{\boldsymbol{\pi}}$ is strictly less than following precommitment path $\boldsymbol{\pi}^{*0}$, and (iii) initialization to TIC, precommitment policy (that is necessary to invoke the insufficiency of standard PI as illustrated in Example 4.6). For our stochastic (S) example, we inject noise to the deterministic transitions $p_9^a(\cdot)$ of state 9 in (D) such that

$$\forall a \in \{\leftarrow, \uparrow, \rightarrow, \downarrow\}, \ P(s'|s = 9, a) = \begin{cases} .9, & \text{if } p_9^a(s') = 1 \text{ in (D)} \\ \frac{1 - .9}{3}, & \text{else} \end{cases}$$

## D.2  Benchmark Algorithms

Following, we describe the two benchmark algorithms that we use in our experiments: `MC` and `sophEU`. For our `MC` implementation, we use the fist-visit variant on-policy MC control[6] as described in Algorithm 2. For the `sophEU`, we adapt the `sophEU` algorithm proposed in Evans et al. (2016) by modifying the exploration technique to $\epsilon$-greedy; see Algorithm 3. This is done for fair comparison with the other two methods, i.e. `MC` and `bwdQ`.

## D.3  Training Setup

For each pair of algorithm and environment, hyperparameters are informally selected from the sets $\alpha, \alpha_Q \in \{.2, .3, .4, .5\}, \alpha_r \in \{.7, .8, .9, 1.0\}, \epsilon \in \{.01, .03, .05, .07, .1\}$ with the following criteria in mind: (i) small overtaking-mean $i_{21}^*$, (ii) small stdev-shade on the Q-value learning curves at $s = 9, 21$, and (iii) identifiable $i_9^*, i_{21}^*$ (i.e. reducing the overlapping frequencies between two contending actions' mean Q-value learning

---

[6]We refer to the sourcecode in https://github.com/dennybritz/reinforcement-learning prior to our hyperbolic-discounting modification.

---

**Algorithm 3** Sophisticated Expected-Utility Agent (sophEU)

---

1: **Input:** Hyperbolic ($k = 1$) Gridworld, Hyperparameters($\epsilon, \bar{T}, \alpha$)
2: **Output:** Approximate SPE Q-function $Q^{\hat{\pi}}(s, a) = Q(s, a, 0), \forall s \in \mathcal{S}, a \in \mathcal{A}$
3: **Initialize:** $Q(s, a, d) \leftarrow 0, \forall d, s \in \mathcal{S} \setminus \bar{\mathcal{S}}, a \in \mathcal{A}$; $Q(s, a, d) \leftarrow \varphi(0)R(s, a), \forall d, s \in \bar{\mathcal{S}}, a \in \mathcal{A}$; $\pi'_d(s) \leftarrow$
   $\arg\max_a Q(s, a, d), \forall d, s \in \mathcal{S}, \boldsymbol{\pi} \leftarrow \emptyset$
4: **repeat**
5:    Update $\boldsymbol{\pi} \leftarrow \boldsymbol{\pi}'$
6:    Choose $S_0$ randomly
7:    **for** $t \leftarrow 0$ to $\bar{T} - 1$ **do**
8:       Sample action $A_t \sim \pi_0^\epsilon(.|S_t)$
9:       Observe reward $R_{t+1} \doteq R(S_t, A_t)$ and next state $S_{t+1}$
10:      Set $d \leftarrow t$
11:      Compute utility $U \leftarrow \varphi(d) \cdot R(S_t, A_t)$
12:      Compute expectation $E \leftarrow \sum_{a' \sim \mathcal{A}} \pi_0^\epsilon(a'|S_{t+1})Q(S_{t+1}, a', d + 1)$
13:      Update $Q(S_t, A_t, d) \leftarrow Q(S_t, A_t, d) + \alpha(U + E - Q(S_t, A_t, d))$
14:      Update $\pi'_d(S_t) \leftarrow \arg\max_a Q(S_t, a, d)$
15:   **end for**
16: **until** stable ($\boldsymbol{\pi}'_0 \neq \boldsymbol{\pi}_0$)

---

curves); see Figure 8(a)-(b) for relatively bad instances. For all environments and algorithms, we set $\bar{T} = 100$; larger episode truncation does not affect much our experiment results. We summarize our final choice of hyperparameters in Table 2.

Table 2: Hyperparameters

| $(\epsilon, \bar{T}, \alpha_Q/\alpha, \alpha_r)$ | MC | sophEU | bwdQ |
|---|---|---|---|
| (D) | (.07, 100, -, -) | (.07, 100, .4, -) | (.07, 100, .4, 1.0) |
| (S) | (.07, 100, -, -) | (.07, 100, .4, -) | (.07, 100, .4, .9) |

## D.4 Additional Results and Evaluation

This subsection expands the results and evaluation subsection in the main paper.

### D.4.1 Q-value Learning Curves

To illustrate how we record the overtaking indexes $i_9^*, i_{21}^*$ used to compute $\Delta i^*$ in Table 1, we plot in Figure 5-8 the Q-value learning curves that correspond to Figure 4.

### D.4.2 Terminal Policies vs Groundtruth Value Comparisons

In Figure 4(b) of the main paper, we have shown that all algorithms will eventually terminate at SPE policy $\hat{\pi}(s_0)$ for $s_0 = 21$. However, Figure 4(a) shows that both MC and sophEU do not flatten to the groundtruth SPE value function $V^{\hat{\pi}}(s_0) = Q^{\hat{\pi}}(s_0, \hat{\pi}(s_0))$. Now that we have Q-value learning curves in Figure 5-8, it becomes clearer that the source of this discrepancy lies on the mis-evaluated Q-values; see $Q(21, \rightarrow)$ in Figure 5(a) for instance. This is explainable for a few reasons. Firstly, in the case of MC, the magnitude of exploratory rate $\epsilon$ causes Q-values to evaluate the exploratory policy $\boldsymbol{\pi}^\epsilon$ consisting paths of extended lengths, which correspondingly lead to an underestimated cumulative discounted reward. In the case of sophEU, similar undervaluation of $\boldsymbol{\pi}$ happens due to the action-taking probabilities being included in the Q-table updates; see line 12-13 in Algorithm 3. While making $\epsilon$ smaller intuitively fixes this issue, learning performance deteriorates (i.e. highly variable across seeds) once we decrease $\epsilon$ up to certain threshold; our final choice of $\epsilon = .07$ has taken this into consideration. Secondly, MC observes some kind of smoothening effect across updates, which if combined with the delayed reflecting of information (i.e. prolonged $\Delta i^*$) exacerbate the

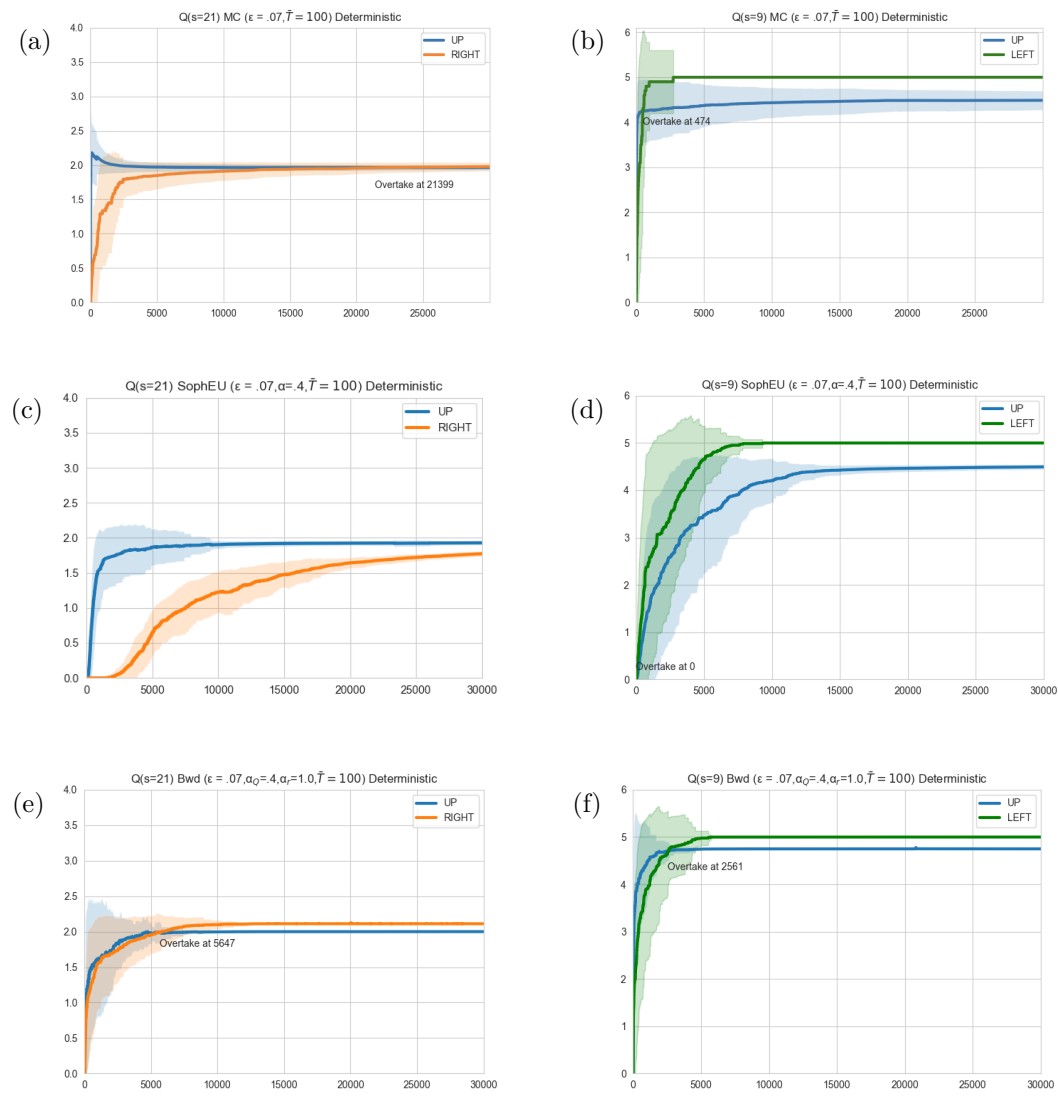

Figure 5: Q-value Learning Curves for MC, sophEU, and bwdQS at $s = 21, 9$ in (D).

early flattening of policy values. Such smoothening concurrently explains how MC appears to have lesser variance as compared to bwdQ or sophEU at later iterations; see Figure 4(a)-Stochastic (S) in the main paper.

### D.4.3 Ablation Study: Reversed Backward Q-learning

Since both benchmark algorithms suffer from similar undervaluation of policy issue, we construct an additional benchmark: Reversed Backward Q-learning (bwdQ-rev), that is based on our own algorithm bwdQ. Here, we only retain the extended DP-based policy evaluation component of bwdQ (that resembles TD-based methods in standard RL literature) and apply standard conditioning by reversing the backward order of policy update in line 11, Algorithm 1. This benchmarking can also be seen as an ablation study to see how backward conditioning alone can reduce delusionality and improve learning performance. Figure 9 displays the value and Q-value learning curves of bwdQ-rev against bwdQ in both (D) and (S), under the same learning rates.

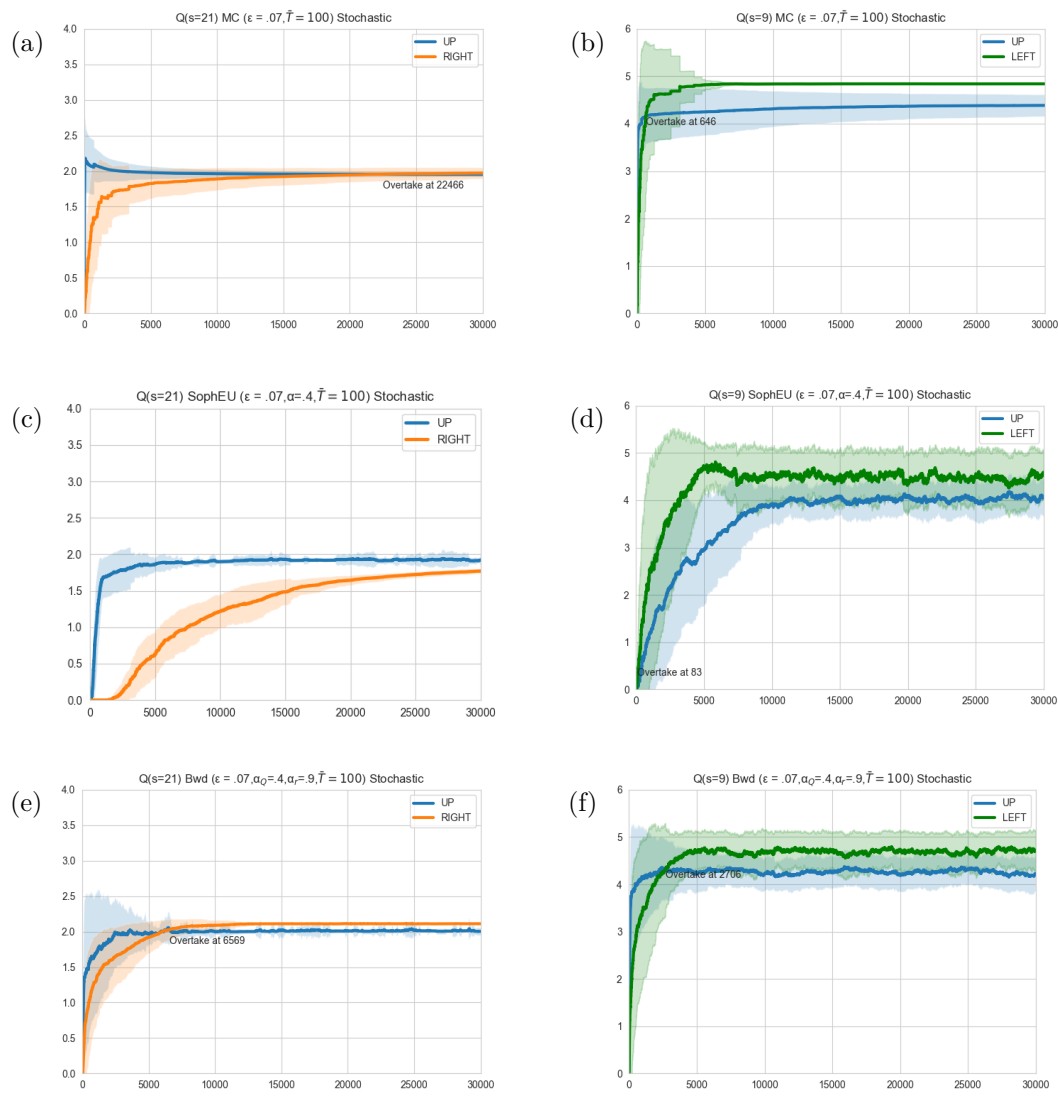

Figure 6: Q-value Learning Curves for `MC`, `sophEU`, and `bwdQ` at $s = 21, 9$ in (S).

**Value learning curves.** From Figure 9(a), it can be seen that `bwdQ-rev` also manages to match the groundtruth values at about the same speed of `bwdQ`. However, we can observe a large swing at earlier iterations which indicate `bwdQ-rev`'s degree of delusionality. In particular, such a swing is caused by 21's late update about 9's strategy of going '←', resulting in delusional prediction targets $Q(21, \uparrow; \pi(9) = \uparrow)$ and an inflation of Q-values. `BwdQ-rev`'s speed of correction towards the groundtruth here is then made possible by its large learning rates[7], which we will show to have some disadvantages next.

---

[7]Some comparisons can be made with `MC`'s degree of delusionality in Figure 4(a) of the main paper, that is milder for its smaller (smoothened) learning rate. It is then natural to ask how `sophEU` does not seem to exhibit such (Q-)value inflation. This can be explained by `sophEU`'s *delay-augmentation*, in which the rate of value propagation from delays $d > 0$ to $d = 0$ may match the speed of delusionality correction. To illustrate, we can observe how in Figure 5(c), `sophEU`'s $Q(21, \uparrow)$ climbs up slowly from 0 instead of jumping to near 2.0 like most others.

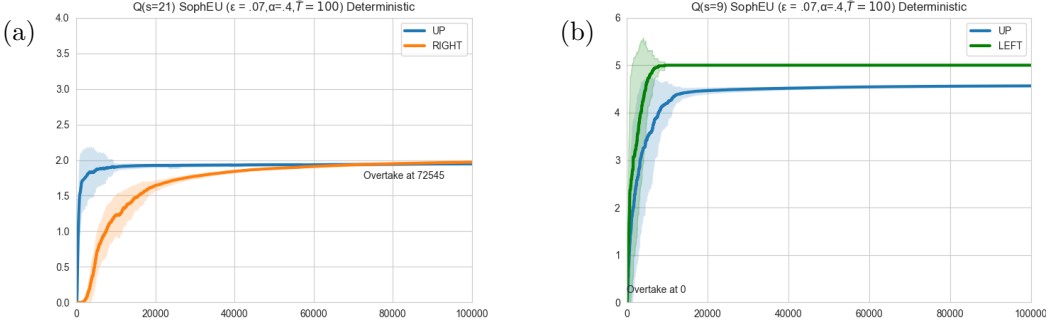

Figure 7: Q-value Learning Curves for `sophEU` (Ext.) at $s = 21, 9$ in (D).

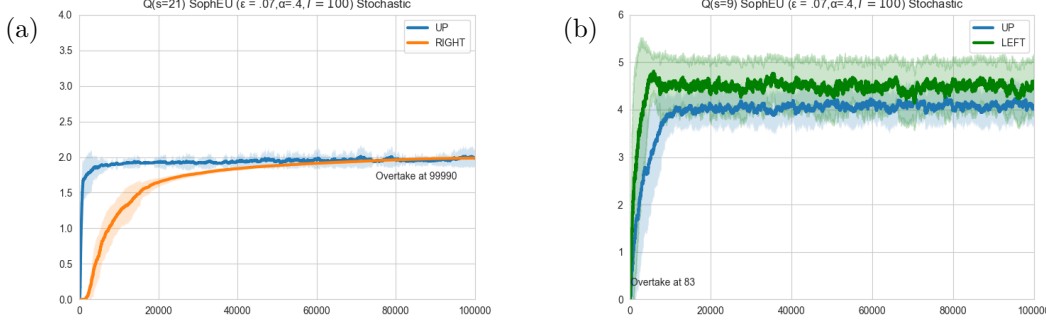

Figure 8: Q-value Learning Curves for `sophEU` (Ext.) at $s = 21, 9$ in (S).

**Q-value learning curves.** From Figure 9(e), while $i_{21}^*$ of `bwdQ-rev` seems to match `bwdQ`, we observe wide stdev-shades for two contending actions '→' and '↑' that overlap throughout training episodes, indicating indecisive behaviour i.e. high variability of trained policy at convergence across random seeds. We note that this phenomenon happens in 5 out of 10 `bwdQ-rev` experiments we conducted under (S) setup, while never happening in `bwdQ`. Moreover, in Figure 9(f), the mean Q-curves of the two contending actions '←' and '↑' are relatively unstable and overlap frequently; see how `bwdQ` behaves in Figure 6(f) for comparison. These evidences suggest that reversing backward conditioning to standard impedes learning, particularly impairing `bwdQ`'s ability to handle larger learning rates. The results for both algorithms under (D) setup are largely similar, except for `bwdQ-rev`'s inflated Q-values at earlier iterations that has been covered previously.

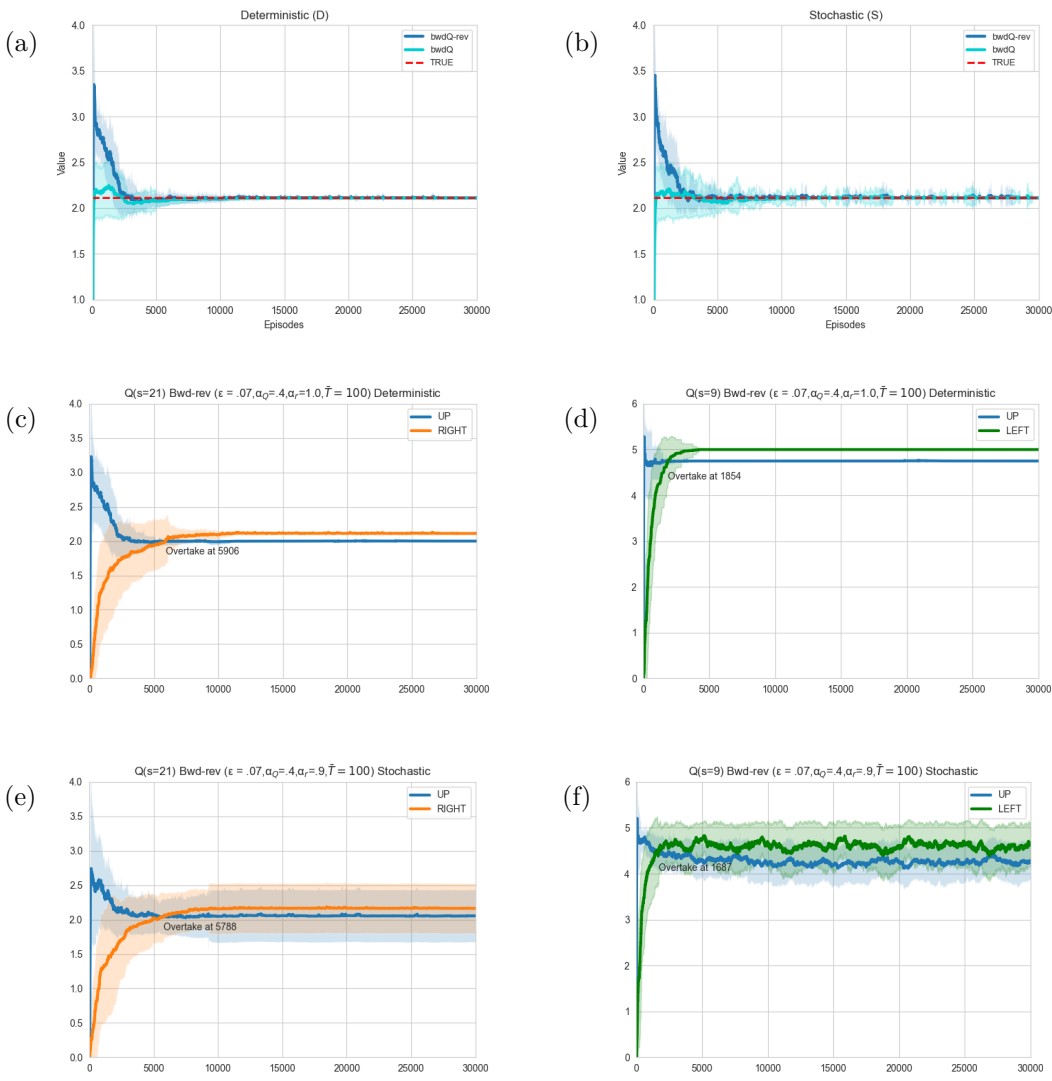

Figure 9: **Value and Q-value Learning Curve at** $s = 21, 9$ **of** `bwdQ-rev` **in (D) and (S).** Experiment ID for `bwdQ-rev` is similarly set to the one exhibiting slowest termination at 21 as indicated by the largest $i_{21}^*$.

