# OpenReview forum: "Reinventing Policy Iteration under Time Inconsistency"
_TMLR — Accepted by TMLR_

### Review · Reviewer_agjL · 2022-08-11

**Summary Of Contributions:**

The paper tackles the Reinforcement Learning (RL) problem in the presence of time-inconsistent (TIC) objective functions, i.e., the case in which the usual Bellman optimality principle is violated (e.g., hyperbolic discounting instead of exponential one). The goal of the paper is to extend Policy Iteration (PI) to this setting and eventually propose a sample-based approach named backward Q-learning (bwdQ). To this end, the notion of optimality is rephrased in terms of the Subgame Perfect Equilibrium (SPE). Finally, a numerical validation on a grid world environment is proposed in comparison with some baselines.

**Broader Impact Concerns:**

None.

**Requested Changes:**

**Critical**
Addressing the concerns related to [Subgame Perfect Equilibrium], [Assumptions], [Theorem 3.4], [Experimental Evaluation].

**Non-Critical**
Fixing minor issues.

**Strengths And Weaknesses:**

**Strengths**

* [Setting] The paper addresses a setting, the one of TIC objectives, that is of interest in my opinion and seems motivated by the observation of the human behavior that "tends to deviate from the plan".

* [Novelty] The proposed approach is novel, in particular, the employed updated rules for defining backward Q-learning (bwdQ).

**Weaknesses**

* [Subgame Perfect Equilibrium] I have some concerns about the role of the SPE in this context. First of all, SPE is a solution concept from game theory, and, at a general level, I do not understand why this is actually employed in (what I believe is) a single-agent problem. I have to admit that I am not an expert on the TIC RL literature, and, therefore, I might be missing some central points. Furthermore, if I look at Definition 3.1, the inequality is basically requiring that the current policy $\hat{\pi}$ has non-positive advantage, i.e., in the standard RL language, that $\hat{\pi}$ is the optimal policy. Can the authors elaborate on the role of SPE? Nevertheless, my feeling is that the authors could have made a better effort in trying to introduce the role of SPE in TIC RL, even for a non-expert reader.

* [Assumptions] As the authors aknowledge, Assumptions 3.2, 3.3, and 3.4 are technical. The first two are briefly explained and their meaning is clear, while Assumption 3.4 remains obscure to me. I would appreciate it if the authors could provide a more detailed explanation of the meaning of these assumptions. Furthermore, and most importantly, are these assumptions reasonable in practice? For instance, are these assumptions verified in the grid world example?

* [Theorem 3.4] I had trouble evaluating the meaning of Equation (14). What does the approximation symbol exactly mean? Are you approximating and ignoring any term in the expression? In my opinion, this should be clarified.

* [Experimental Evaluation] The experimental evaluation is limited since it is conducted in just two versions (stochastic and deterministic) of the same grid world. I do not see this necessarily as a problem since the main purpose of the paper is to provide a theoretical contribution. Nevertheless, I think that the discussion in Section 6 could be improved. In particular, the results of the experimental evaluation are described in a very synthetic way. For instance, it is interesting to observe that the proposed approach is able to converge to the true value function while the others display a bias. In my opinion, the authors could elaborate on why this happens. Why do the baselines display a bias? Furthermore, I couldn't find the number of runs for generating the plots in Figure 4 and whether the shaded areas represent standard deviation or confidence intervals (if so, with which confidence?).

**Minor Issues**
* The acronym SPE appears in the abstract, but it is never defined before.
* In the introduction, the authors say that "Standard PI has been the basis of many classical RL algorithms, such as the popular Q-learning". In my view, Q-learning can be thought of as the sample-based version of Value Iteration rather than PI.
* Assumption 3.2, Equation (3) should contain an absolute value, not a norm. The same is in Assumption 3.4, for the second line after Equation (4).
* Sometimes, in full-line equations, the punctuation is missing.
* The plots in Figure 4 (a) are non-vectorial.
* [Notation] The used notation seems quite non-standard and not immediate to understand. In particular, I am referring to the notation $\pi_1^T \cdot \pi_2$ to denote the execution of policy $\pi_1$ for the first $T$ steps and then $\pi_2$ (like in Equation (3)). This is not really a weakness of the work, in my opinion, but it seems to me that everything could be formulated in terms of non-Markovian policies.

---

> ### Author Response · Authors · 2022-10-06
> **Response for Reviewer agjL (1)**
>
> Dear Reviewer agjL,
>
> Thank you for your constructive comments and for acknowledging the value of our contributions! We have revised our manuscript to address some of your concerns, which we will indicate below together with our response.
>
> ## SPE
> [SPE on a single-agent problem] Once BPO (i.e. Eq (1)) is violated, $\boldsymbol\pi^{*0} \neq \boldsymbol\pi^{*\tau}$ for some $\tau$ and we will have a collection of *competing optimization problems* $\{\mathcal{P}_{\tau, s_\tau}: \forall \tau \in [0, \infty), s_\tau \in \mathcal{S}\}$
>
> to solve. One can either (i) focus on only a single agent corresponding to $(0,s_0)$, denoted by Agent-$s_0$, and thus solve for a globally optimal or so-called precommitment policy (but it is expensive to obtain in general); or alternatively, (ii) consider multiple agents indexed by $(\tau,s_\tau)$ for $\tau\in[0,\infty)$, where each Agent-$s_\tau$ is associated with the problem $\mathcal{P}_{\tau, s_\tau}$, and solve for the Nash equilibrium of this (sub-)game, i.e. SPE.
>
> > Revision: page 4, Section 3.2.
>
> [Non-positive advantage] As you stated, the policies that realize non-positive advantage in standard RL are equivalent to the optimal policy. We note however that such equivalence draws on BPO, which does not apply in TIC RL. Once BPO is violated, the optimal policy (one that solves $ \mathcal{P}_{0, s_0} \coloneqq  max_\boldsymbol\pi V^{\boldsymbol\pi}(s_0)$
>
> that we refer to as globally-optimal/precommitment policy $\boldsymbol\pi^{*0}$) is no longer equivalent to the one that realizes non-positive advantage (one that solves the game induced by competing local optimizations $\mathcal{P}_{\tau, s_\tau}\coloneqq max_\boldsymbol\pi V^{\boldsymbol\pi}(s_\tau),$
>
> that we refer to as SPE policy $\hat{\boldsymbol\pi}$). For instance, in our Hyperbolic Gridworld, SPE policy (cf. Figure 1(c)) is not equal to the precommitment policy (cf. Figure 1(b)).
>
> > Revision: page 5, Section 3.2.
>
> ## Assumptions
> We would like to first remark that Assumption 3.5 is reasonable in practice for we have shown the existence of an MDP class that satisfies it; please refer to Lemma A.3 for detailed proof and the paragraph after that shows the Hyperbolic Gridworld in Figure 1(a) belonging to this class. We have revised our exposition to clarify the reasoning behind this assumption.
>
> > Revision: page 5, Remark 3.2.
>
> ## Theorem 4.3
> Following your suggestion, we have revised our manuscript to clarify the notation "$\approx$".
>
> > Revision: page 8, Remark 4.4.
>
> ## Experimental Evaluation
> [Results, analysis] We do have more elaborate discussions on our experiment results, which include your concern on biases (in Appendix D.4.2). To summarize, bias in sophEU is introduced due to the use of a stochastic policy. While in MC, bias is induced by the exploration rate that affects MC targets for training $Q$-function. We also supplement more empirical results such as the ablation study of backward-conditioning in Appendix D.4.3. In our current and revised version, these were omitted from the main content as we wanted to focus more on our theoretical findings and meet the space limit for a short submission. We will move these to the main content if deemed necessary.
>
> [Figure 4] To obtain the mean (line) and standard deviation (shade) in Figure 4, we run 50 times the same experiments. In one experiment, we have only one line with values $Q^i(s0, \pi^i(s0)): 1 \leq i \leq 20000$, where $Q^i$ is the learnt Q-function at episode $i$ and $\pi^i(\cdot) = \arg\max_a Q^i(\cdot, a)$. We have revised Figure 4's caption to clarify this.
>
> > Revision: page 13, Figure 4/caption.
>
> ## Minor Issues
> [SPE acronym]
>
> > Revision: page 1, abstract.
>
> [Intro: value iteration] We agree that value iteration should connect policy iteration to Q-learning and have revised accordingly.
>
> > Revision: page 1, intro.
>
> [Assumption 3.2, norm] Thank you for pointing this out! We have revised all appearances of the norm in our manuscript.
>
> [Notation] Our current use of notation is due to our SPE definition that builds on the Markovian property. Moreover, we would like to remark that to study the case of non-Markovian policies, we may need to introduce a different kind of SPE policy such as the one used in [Latt14] that is beyond our current scope.
>
> ## References
>
> [Latt14] Lattimore, T. and Hutter, M., 2014. General time consistent discounting. Theoretical Computer Science, 519, pp.140-154.

---

> > ### Comment · Reviewer_agjL · 2022-10-31
> > **Reply to Authors Response**
> >
> > I thank the authors for having provided clarifications about the concerns I raised and for the amendments to the paper. I think now the paper provides a relevant contribution and, thus, I recommend acceptance.

---

> > > ### Author Response · Authors · 2022-11-01
> > > **Thank You**
> > >
> > > Dear Reviewer agjL,
> > >
> > > Thank you for the positive feedback!

---

### Review · Reviewer_PRKT · 2022-08-23

**Summary Of Contributions:**

The submission tackles the problem of optimizing a policy under time inconsistency*: *i.e.* when decision optimality varies across time. This happens when the objective function (which devises optimality) is not expressible as the standard expectation of the geometrically discounted sum of rewards: $\mathcal{J}(\pi)\doteq \sum_t \gamma^t R_t$. The Bellman's principle of optimality does not hold anymore which compromises the policy iteration monotonous improvement. The authors derive an alternative optimality definition called the subgame perfect equilibrium. They analyze policy iteration under this optimality criterion and propose an algorithm called backward Q-learning. Finally, they validate their work on two small gridworld environment.

\* Example 3.6 is quite informative on the phenomenon and allows also to relate to human decisions. I believe it should be presented as soon as the introduction. Indeed, the gridworld is like choosing to follow a longer path to avoid the temptation of buying croissants on the way to work. It also reminds me of a friend that uninstalled video games in order not to be tempted to play them: you make sure that the distractions never yield immediate reward, you need to plan to access them. These examples show a lack of personal focus if you ask me, but it is clear that we all regularly succumb to immediate distractions of our long term goals. For these reasons, I am quite hyped by the subject of research.

**Broader Impact Concerns:**

I do not have impact concerns.

**Requested Changes:**

I request two things from the authors:
1- solve the exposition issues
2- answer my technical concerns

**Strengths And Weaknesses:**

Strengths:
* as I said in the introduction, I love the problem at stake and I would encourage the authors to continue on this topic.
* the work seems well-rounded: all the elements I would expect from an article are present. Obviously more ambitious experiments would be a plus, but this is not a requirement in my eyes.

Weaknesses:
* The exposition mainly, under various forms:
  * concepts are often not properly defined:
    * abstract: *time inconsistent (TIC) objective* => the authors introduce PI, but do not explain what TIC is precisely.
    * introduction: *monotonicity* => equivalent terminology is provided, but the concept is not defined.
    * introduction: *termination policy* => I am still unsure what it is.
  * acronyms are sometimes not introduced:
    * abstract: *SPE* => we do not understand it refers to subgame perfect equilibrium at this point.
    * introduction: *PIT* => I thought it was a typo for PI, but it's actually PI theorem. Maybe this acronym is not necessary.
    * introduction: *DP* => not introduced.
    * section 3.1: *BPO* => recall what it stands for.
  * some English language needs proof reading:
    * abstract: *subgame perfect equilibrium* => there is a sentence trying to explain it but it is not understandable.
    * section 2: *problems are computationally intractable such that new tractable solutions are required* => if a problem is intractable, it means that we know that there is no tractable solution. I don't understand what the authors mean here.
    * section 3.2: *when initiated at any intermediate states $s_t$ against $s_0$* => I don't understand.
  * some mathematical imprecisions:
    * definition 3.1: the action space has been assumed to be countable earlier, which somehow does not guarantee the existence of an SPE policy. Indeed, consider a stateless MDP that grants a reward of $1-\frac{1}{a}$ with $a\in\mathcal{A}=\mathbb{N}$.
    * definition 4.2: $m$ was previously defined as $t-\tau$, so it should be from $0$ to $T-\tau$ in this definition. The same remark applies many times again after.
    * equation 11: $\phi$ was defined to be in $[0,1)$, if it's exactly equal to 0, then (11) is not defined.
    * equation 13: it is incomplete. Therefore, I don't know where the terms in eq (14) come from.
  * the authors do not really help the reader to follow the train of thought:
    * Assumptions should be names so that we understand in a glimpse what they intend to say.
    * Assumption 3.4: it is not explained. What does it say and how constraining it is? I have doubts about this one, see my technical concerns.
    * Assumption 3.4: it's a recursive definition, usually the initial definition (for t=0) is written before the recursion. It helps the reader to parse the definition.
* I also have a few technical concerns which I'll ask the authors to address:
  * Assumption 3.4 seems to defeat a large part of the impact of the work. Indeed, it looks like this assumption enforces a discounting that is below-geometric. In particular, I don't think it works with the hyperbolic discounting, and Lemma A.3 seems to confirm my intuition.
  * Section 3.3: *general discounting function* => is this work limited to general discounting function, because it seems to claim that it can be any TIC? Does not this discounting need to be monotonously decreasing?
  * What about the solution used by Fedus consisting of using a family of discount factor as basis functions for any discounting scheme \phi? It would only work for diminishing discount factors, but that looks powerful to me, more than the approximate solution proposed by the authors. So I would like to hear their arguments against this approach.

---

> ### Author Response · Authors · 2022-10-06
> **Response for Reviewer PRKT (1)**
>
> Dear Reviewer PRKT,
>
> Thank you for your detailed reviews and for acknowledging the value of our contributions! We have revised our manuscript following your suggestions to address the exposition issues; each of which we will annotate below together with our response to your technical concerns.
>
> ## Exposition
> ### Concepts are not properly defined
> [abstract: TIC]
>
> > Revision: page 1, abstract.
>
> [intro: monotonicity]
>
> > Revision: page 1, intro/footnote 2.
>
> [intro: termination policy]
>
> > Revision: page 2, first line.
>
> ##
>
> ### Acronyms are sometimes not introduced
> [abstract: SPE]
>
> > Revision: page 1, abstract.
>
> [intro: PIT]
>
> > Revision: page 2, intro/contribution.
>
> [intro: DP]
>
> > Revision: page 1, intro/footnote 1.
>
> [Section 3.1: BPO]
>
> > Revision: page 4, Section 3.1.
>
> ##
>
> ### Some English language needs proofreading
> [abstract: SPE] As there is limited space in the abstract, we decide to instead explain the connection between SPE and TIC in the Intro. We would like to also remark that defining TIC and SPE in more detail than the current revised Intro, including inserting Example 3.7, might require us to introduce our notations earlier (which may not be ideal in our opinion). Thus, we use Section 3.2 to cross-reference back to the more intuitive Intro. Do let us know if you have more suggestions on this issue!
>
> > Revision: page 1, abstract; page 1-2, intro/paragraph 2, 4; page 5, paragraph 1.
>
> [related works: intractable]
>
> > Revision: page 2-3, related works/paragraph 1-2.
>
> [Section 3.2: Assumption 3.5 explanation]
>
> > Revision: page 5, below Assumption 3.5.
>
> ##
>
> ### Some mathematical imprecisions
> [Definition 4.2: $m$ inconsistently defined]
>
> > Revision: page 6, Definition 3.6.
>
> [Equation 11: division by 0]
>
> > Revision: page 6, 1st line.
>
> [Equation 13: incomplete Q-function]
>
> > Revision: page 8, Theorem 4.3.
>
> [Definition 3.1: countable action-space] Thank you for pointing this out! We have revised it accordingly. To better clarify our setup, we would like to remark that stateless MDPs are excluded from our setup because then we are not able to form a game (whose players are states).
>
> > Revision: page 3, Section 3.1.
>
> ##
>
> ### Authors do not really help readers to follow the train of thought
> [Assumption 3.5: recursive definition, explanation] To address your concerns, we have revised our manuscript to clarify the intuition behind such an assumption and make the cross-reference to Appendix A.1 more visible. We defer our response to your specific technical concerns below.
>
> > Revision: page 5, Assumption 3.5; below Assumption 3.5.
>
> ## Technical Concerns
> ### Assumption 3.5: below-geometric discounting, Lemma A.3.
> To address your concern, we will show that some discounting functions faster or equal to geometric do satisfy condition (24) in Lemma A.3. For instance, given $\varphi(t) = \gamma^{kt}$, we have $\frac{\varphi(\tau + t)}{\varphi(\tau)} = \gamma^{kt}$ that is a constant function of $\tau \Rightarrow$ increasing in $\tau$. Or given hyperbolic discounting $\varphi(t) = \frac{1}{1 + k(t)}$, we have for any $k \geq 0$:
> $$\frac{d}{d\tau}\frac{\varphi(\tau + t)}{\varphi(\tau)} = \frac{d}{d\tau}\frac{1+k(\tau)}{1+k(\tau+t)} = \frac{k(1 + k(\tau + t)) - k(1 + k\tau)}{(1 + k(\tau + t))^2} \geq 0$$
> Thus, $\frac{\varphi(\tau + t)}{\varphi(\tau)}$ is increasing in $\tau$.

---

> > ### Author Response · Authors · 2022-10-06
> > **Response for Reviewer PRKT (2)**
> >
> > ### Section 3.3: general-discounting or all TIC
> > To address your concerns on the generality of our work, we will label our main results based on whether they only apply to general-discounting or generalize to all TIC:
> >
> > - Section 3.1: infinite-horizon TIC formalism as BPO violation (**all TIC**)
> > - Section 3.2: infinite-horizon SPE formalism (**all TIC**)
> > - Section 3.2: Assumption 3.3-3.5 (**all TIC**);
> >   >its sufficiency conditions in Appendix A (**general-discounting**), its usage on Theorem 4.3 (**general-discounting**), its usage on Proposition 5.3 (**all TIC**)
> >
> > - Section 3.3: Counterexample of BPO (**general-discounting**)
> > - Section 4.1: SPE optimality of standard PI (**all TIC**)
> >  - Section 4.2 (**general-discounting**)
> > - Section 4.3: a counterexample to update monotonicity (**general-discounting**), everything else (**all TIC**)
> > - Section 5.1 (**all TIC**).
> >   >  Here, we would like to remark that despite our general-discounting illustration in both Example 4.6 and 5.4, we can entirely reuse our arguments as long as we have identified at least 2 states exhibiting TIC and their optimal *delayed* and *undelayed* actions, both of which happen (in the construction of BPO counterexample) independently of our arguments.
> >
> >  - Section 5.2: Approximate Backward Conditioning/Algo 1's TD evaluation in l.12-17 (**general-discounting**), backward update in l.11, 18-20 (**all TIC**)
> > - Section 6: Experiment results and analysis (**general-discounting**)
> > - Appendices (**general-discounting**)
> >
> > To summarize, any results that build directly on $Q$-function or value function without the need to specify their definitions extend beyond general discounting. Thus as listed above, only those related to TD evaluation (for deriving recursion needs the definition of $Q$-function), counterexamples, and experimental results are limited to general discounting.
> >
> > As for how restrictive our general-discounting is, as you stated, condition (24) imposes monotonically decreasing $\varphi: \mathbb{N} \rightarrow (0, 1]$. However, we would like to remark that such a condition is common since a discounting factor is expected to be decreasing. Moreover, (24) is only a sufficient condition for Assumption 3.5.
> >
> > ### Comparison with [Fedus19]
> > We elaborate on the difference between ours and [Fedus19]'s approach in 3 ways:
> >
> > [i] As you stated, [Fedus19] and ours have different approximation schemes. While ours lies in the TD evaluation (i.e. Eq(13)), [Fedus19] uses their Eq(25)-(26):
> > > $$\sum_{\gamma_i \in \mathcal{G}} (\gamma_{i+1} - \gamma_i) w(\gamma_i) Q^{\gamma}_{\boldsymbol\pi}(s, a) \approx \int_{\gamma \in (0, 1)} w(\gamma)Q^{\gamma}_{\boldsymbol\pi}(s, a) d\gamma$$
> >
> > where $\mathcal{G}$ is a partition of $(0,1)$. The restrictions on our MDP arise from the need to *theoretically* bound our *Q-function*approximation error. In contrast, [Fedus19] focuses on how well their scheme approximates the *discounting function empirically* (in their Appendix F).

---

> > > ### Author Response · Authors · 2022-10-06
> > > **Response for Reviewer PRKT (3)**
> > >
> > > [ii] In practice, [Fedus19] approximation leads to the need to tune $\mathcal{G}$, both its size and element locations $\gamma_i \in (0, 1), \forall i$. This could be both expensive and difficult to tune. For instance, even in our easier environments, [Fedus19] needs $|\mathcal{G}|=1000$ to perform (note that in [Fedus19], they consider $|\mathcal{G}|=510$ only for their experiments).
> > >
> > > > To better illustrate, please refer to: [Figure: Fedus200 vs Fedus100 vs bwdQ in (D) and (S)](https://drive.google.com/file/d/1TsnUlitW4Jf-oEUimME1aW167fDSMYtI/view?usp=sharing)
> > >
> > > By this, we mean that the bias from smaller $|\mathcal{G}|=200$ causes inaccuracy in the final action choice (i.e. $\uparrow$ chosen by $s = 21$ when $\hat{\pi}(21) = \rightarrow$), while with $|\mathcal{G}|=1000$, the algorithm (in deterministic (D) environment) successfully outputs the correct actions despite persisting bias (their asymptotic bias can disappear only when $|\mathcal{G}|=\infty$). We remark however that the above observations are based on \textit{uniform} locations of $\gamma_i, \forall i$, as following their suggestions on training with larger $\gamma$'s (in their Appendix C) did not seem to improve the results of our reimplementation. In contrast, our scheme only needs the initial specification of $\bar{T}$; informed by our uniform approximation error bound (i.e. Eq (13)) across state-action-policies, $\bar{T}$ can be chosen to preserve the correctness of final action choices. Another advantage of our $\bar{T}$ is its ability to self-adjust during training, e.g. even when we set $\bar{T} = 100$, most sampled trajectories terminate (i.e. reach goal states) at $T^* \leq 10$. This can reduce the computational burden (e.g. ours: $\frac{T^*(T^*+1)}{2} \approx 55$, [Fedus19]: $|\mathcal{G}| = 1000$), which resulted in significantly shorter running time (e.g., ours: $\sim 10$s, [Fedus19]: $\gtrsim 370$s).
> > >
> > > We admit that such results may depend on the environments (say, the existence of absorbing goal states) and function approximations may alleviate the action-dependent bias in [Fedus19] (thus, mitigates inaccuracy in final action choices). In this sense, as you pointed out, [Fedus19] may be more powerful as it has been tested on more complex domains. We agree that such scaling up is an important property. However, from our current paper, it is non-trivial to extend the results theoretically to more complex domains (e.g., higher dimensions, games), where we need to address the issues of training instability as well as the related theory of SPE. We have put these in our research agenda.
> > >
> > > [iii] As you pointed out, Lemma A.3's condition (24) does limit the generality on our discounting function $\varphi(.)$. We note however that restrictions on $\varphi(.)$ are also present in [Fedus19]'s Lemma 5.1:
> > > > $$\exists w: [0, 1] \rightarrow \mathbb{R} \text{ s.t. } \varphi(t) = \int_0^1 w(\gamma)\gamma^t d\gamma$$
> > >
> > > These are different sets of restrictions and may not be directly comparable (whether one subsumes the other). It is easy to see that their examples of discounting functions (in their Appendix B) fall under our condition (24): $\frac{d}{d\tau}\frac{\varphi(\tau + t)}{\varphi(\tau)} \geq 0$, even for the one we did not consider, i.e., $\varphi(t) = \frac{1 - e^{-kt}}{kt}$. However, their restrictions seem to exclude the discounting function $\varphi(t) = \frac{1}{1 + kt^2}$ while ours does not.
> > >
> > > ## References
> > > [Fedus19] Fedus, W., Gelada, C., Bengio, Y., Bellemare, M.G. and Larochelle, H., 2019. Hyperbolic discounting and learning over multiple horizons. arXiv preprint arXiv:1902.06865.

---

> > > > ### Comment · Reviewer_PRKT · 2022-10-10
> > > > **Feedback on the rebuttal**
> > > >
> > > > I would like to thank the authors for their thorough and honest rebuttal.
> > > >
> > > > I acknowledge my error regarding my first technical concern (Assumption 3.4 enforces a discounting that is **above**-geometric).
> > > >
> > > > Regarding the all-TIC / general discounting nature of the results, I regret that the article is not more homogenous, but this is not a matter of concerns anymore in my eyes.
> > > >
> > > > Finally, I am satisfied with their very complete positioning, comparison and discussion on [Fedus19].
> > > >
> > > > For all these reasons, I will recommend acceptance.

---

> > > > > ### Author Response · Authors · 2022-10-11
> > > > > **Thank you**
> > > > >
> > > > > Dear Reviewer PRKT,
> > > > >
> > > > > Thank you for the positive feedback! As you have encouraged, we will continue working on this research direction toward completing the TIC-related frameworks and analyses.

---

### Review · Reviewer_bU9Q · 2022-09-25

**Summary Of Contributions:**

This paper considers the policy iteration problem for the challenging time-inconsistent (TIC) setting. The paper proposes backward Q-learning (bwdQ), a new algorithm in the approximate PI family that targets SPE policy under general (non-exponential) discounting criteria. They also demonstrate the implications of our theoretical findings on the behavior of the bwdQ and other approximate PI variants via experiments.

**Broader Impact Concerns:**

Not applicable.

**Requested Changes:**

Please see the above section.

**Strengths And Weaknesses:**

I have read the whole paper and part of the proofs. Overall, this paper considers the interesting Time Inconsistent setting and propose backward Q-learning for handling the challenges. I have the following questions for the authors to address.

1. What is the definition of $\bar{T}$ in Assumption 3.2? The missing definition makes it hard to grasp what 3.2 is about.

2. The discounting criterion $(5)$ has $\varphi(\cdot)$ inside and might not be well-defined for $\varphi$ equals to constant $C$. When $\varphi\equiv 1$m shouldn't your result simply give $\infty$?

3. The Assumption 3.4 is quite hard to parse, which makes the result not easy to understand, in particular, why it is reasonable to assume $\lim_{\epsilon\rightarrow 0} \epsilon/\kappa=0$?

4. I am not sure about the Time inconsistency utilized in Algorithm 1. While I can see the TIC-adjusted TD evaluation is used in line 12-17, the hindsight for using backward conditioning remains lacking. Maybe I missed some point, but could you please explain more about the power of the design for Algorithm 1 and why this overcome the hurdle in standard PI?

5. May I ask what is the main technical novelty of Theorem 4.3? It seems you provide a Bellman style equation (14) for the TIC setting, how hader is it compared to the standard Bellman equation besides that the reward uses general discounting?

6. Currently, there is no convergence analysis for Algorithm 1, which seems necessary for theoretical validation of the proposed algorithm. In addition, the work would be more complete if the finite sample regret analysis can be conducted since due to the randomness in the trajectory sampling (Line 10 of bwdQ). Note such a result has been conducted for standard Q learning [1].

[1] Is Q-learning provably efficient? NeurIPS 2018.

7. Lastly, the whole work (especially the experiment) focuses on the Tabular setting (Gridworld), which limits of the power of the proposed algorithm. Could you propose similar algorithm for the general setting like Linear MDPs? Having discussion on those aspects would make the contribution of this paper more clear.

---

> ### Author Response · Authors · 2022-10-06
> **Response for Reviewer bU9Q (1)**
>
> Dear Reviewer bU9Q,
>
> Thank you for your detailed reviews and insightful comments! We have revised our manuscript to address some of your concerns, which we will indicate below together with our response.
>
> ## Q1
> $\bar{T}$ is a truncation step of policy, e.g. $\boldsymbol\pi_1^{\bar{T}}\cdot\boldsymbol\pi_2$ signifies the use of $\boldsymbol\pi_1$ from step $[0, \bar{T})$ and $\boldsymbol\pi_2$ from step $[\bar{T}, \infty)$; this notation is introduced in Section 3.1. In Assumption 3.3, $\bar{T}$ is used to enumerate all truncation steps in $[0, \infty)$ to define $\bar{T}_{s, \epsilon} \coloneqq \min\{\bar{T}: |V^{\boldsymbol\pi}(s) - V^{\boldsymbol\pi^{\bar{T}}\cdot\ddot{\boldsymbol\pi}}(s)| \leq \epsilon\}$.
>
> > Revision: page 5, Assumption 3.3.
>
> ## Q2
> We agree with you that having a constant $\varphi(\cdot)$, without any restrictions imposed on the MDP or value function itself, might lead to unbounded $V^{\boldsymbol\pi}(s)$ for some $s, \boldsymbol\pi$. However, in our work, we have excluded such case by Assumption 3.3 and provides an example of general-discounting MDP that satisfies it; please refer to Appendix A.2, Definition A.1, and Lemma A.2 for more details
> > Revision: page 5, footnote.
>
> ## Q3
> The assumption $\lim_{\epsilon/\downarrow 0}\epsilon/\kappa = 0$ is reasonable as we showed the existence of an MDP class that satisfies it (as a sufficient condition); please refer to Lemma A.3 for a detailed proof and the remark after (page 17, before Appendix A.2) that shows the Hyperbolic Gridworld in Figure 1(a) belonging to this class. Moreover, we have revised our manuscript to clarify the intuition behind such an assumption and make the above points more visible in the main content.
> > Revision: page 5, below Assumption 3.5.
>
> ## Q4
> In Algorithm 1, backward-conditioning can be seen in line 11 that reorders updates from the end of trajectory $T^*$. How it overcomes the hurdles in standard PI is discussed in Sections 4.3 and 5.1, specifically through Examples 4.6 and 5.4. In summary, backward-conditioning mitigates inefficient movement of policies (cf. Figure 2, i.e., standard PI suffers from future deviations arising from TIC) by propagating information about future players' "optimal" policies as soon as it is found to earlier players (cf. Figure 3). To see how this phenomenon transfers to a TIC ''Q-learning'' setting, you may refer to Table 1 in Section 6.1 and Appendix D.4.3, particularly to (i) the reduced delusionality in earlier episodes under backward-conditioning (cf. Figure 6(e)) as compared to standard forwardly-ordered version (cf. Figure 9(e)), and (ii) the improved stability of Figures 6(e)-(f) relative to Figures 9(e)-(f). We would like to remark that our observations are consistent with the results of related literature under non-TIC motivations (e.g., sample efficiency [Lin91c; Lee19], consistent uncertainty propagation [Bai21]), where they have similarly concluded backward update's effectiveness in info propagation.
>
> > Revision: page 11, Remark 5.5; page 11-12, Section 5.2.
>
> ## Q5
> The main technical novelty of Theorem 4.3, relative to standard Bellman equation, is implied by the use of general (non-exponentially) discounted rewards and more importantly, in an infinite-horizon setting.
>
> ##
> [TIC (with non-exponential discounting) and adjustment functions] Due to the rewards being non-exponentially discounted, we need adjustment functions to establish a Bellman-like equation for our $Q$-function both for a policy evaluation (given a fixed policy) and a control version. Moreover, the introduction of the adjustment functions is meaningful as the resulting Bellman-like equation characterizes the SPE (when we view the decision-making as an intrapersonal game).
>
> ##
> [Infinite-horizon TIC, approximation, and problem assumptions] Theorem 4.3 mainly addresses the difficulty of applying such adjustment functions in an infinite-horizon setting (by focusing on policy evaluation): unlike in finite-horizon case, we cannot establish equality for Eq (13). Instead, we aim for an approximation and need to manage the induced error by: (i) quantifying and bounding the error (i.e. $\mathcal{O}\left(\frac{\epsilon}{\kappa(\epsilon)}\right)$ and $\lim_{\epsilon\downarrow 0} \frac{\epsilon}{\kappa(\epsilon)}$ by Appendix B.2); and (ii) identifying the problem features that can guarantee a small error (i.e., conditions on value functions: Assumptions 3.2-3.4; conditions on MDP and discounting function in Appendix A.1).

---

> > ### Author Response · Authors · 2022-10-06
> > **Response for Reviewer bU9Q (2)**
> >
> > ## Q6
> > [Q-learning-like convergence analysis] While we agree with you that convergence analysis is an important theoretical validation, from our current work, such results are not trivial to obtain. In particular, despite the natural construction of Q-learning-like algorithms from Theorem 4.3, the coupling features between $r$-functions and $Q$-function are new to the literature and current convergence analysis techniques are not readily applicable to address this. For instance, standard Q-learning analysis typically builds on stochastic approximations on one sample path ($\{Q_0, Q_1\coloneqq\hat{\mathcal{T}}Q_0, Q_2 \coloneqq \hat{\mathcal{T}}Q_1, \dotso\}$) to show contraction ( $\mathbb{E}[ \hat{\mathcal{T}} Q_i - \hat{\mathcal{T}}Q^* | | \mathcal{F}_i]|| \leq \gamma || Q_i - Q^* || + \epsilon_i$ )
> >
> > in the supremum norm $ ||Q|| \coloneqq \max_{s,a}|Q(s, a)| $. This does not apply to our case with two interleaving updates ($ \{ r_0, Q_0, r_1\coloneqq \tilde{\hat{\mathcal{T}}}(r_0, Q_0), Q_1 \coloneqq \tilde{\hat{\mathcal{T}}}\}(Q_0, r_1), \dotso $). In the above, we have used $\hat{\mathcal{T}}$ and $\tilde{\hat{\mathcal{T}}}$ to denote respectively the Bellman operator in standard Q-learning and the TIC-adjusted Bellman-like operator in our backward Q-learning. There are more recent analyses on coupled iteration, such as double Q-learning [Hasselt10; Xion20; Zhao21] that have exploited the symmetry between their double Q-function estimators, i.e., $Q^A$ and $Q^B$, and some decoupling of error dynamics $\epsilon_i \coloneqq ||Q^A_i - Q^B_i||$ from the dynamics of $Q^A_i$ (or $Q^B_i$), i.e., $\epsilon_i \downarrow 0$ can be shown independently of $Q^A_i$ (or $Q^B_i$). However, such arguments do not apply to our case for the asymmetry between the r-functions and Q-function and the fully-coupled dynamics of $Q_i$ and $r_i$, i.e., $\epsilon_i \coloneqq \Delta r_{t, i} \not\downarrow 0$ and its convergence depends on the convergence of $Q_i$ (through $a^*$) and vice versa. Despite these difficulties, we do not exclude the possibility that backward (TIC adjusted) Q-learning can be theoretically guaranteed to converge (under some conditions). For instance, from our experiments, we observed the contraction of r-functions as Q-function stabilizes on some parts of the state space (e.g., those closer to the goal states). For this, we see the need to introduce a new norm or type of contraction that can capture such phenomenon, as we alluded in Remark 5.6 regarding the infinite-horizon formalism to lex-monotonicity. We have put this in our research agenda.
> >
> > > Revision: page 12, Remark 5.7.
> >
> > ##
> > [Regret analysis] Firstly, we would like to remark our focus on ''how to exploit" rather than ''how to explore" and correspondingly, our TIC-related efficiency (i.e., update monotonicity) focuses on how not to waste any given samples that *possibly* give us information about future players' strategies. That said, we agree with you that addressing exploration efficiency would make our exposition more complete (as if we were to get "bad" samples, they will propagate as quickly as "good" samples do). However, to the best of our knowledge, current regret analysis techniques have built on the standard notion of optimality (that focuses on $s_0$), e.g., in [Jin18],
> >
> > $Regret(K) \coloneqq \sum_{k=1}^K [V^*_1(x_1^k) - V^{\pi_k}_1(x_1^k)]$, with $x_1^k = s_0$ and $V^*_1(\cdot)= V^{\boldsymbol\pi^{*0}}(\cdot)$
> >
> > in our notation. Such regret definition may not be extended readily to our "SPE-optimality" that focuses on all players $x_{1:H}^k = s_{0:\bar{T}-1}$ and it will be interesting to explore this direction in future TIC RL works.
> >
> > ## Q7
> > We agree with you on the limitations of tabular setting. In regards to linear MDP, if you are referring to the linear MDP in [Jin20], we note that an episodic (i.e. finite-horizon) setting is not our current focus. There is however a related work [LP21] that has covered finite-horizon TIC and SPE; they have studied a case with linear quadratic transition (in their Section 5) which we believe is connected to linear MDP. Otherwise, if you are referring to linear function approximation for infinite-horizon Q-function, they can indeed be applied to our algorithm through the minimization of TD-errors arising from the coupled recursion of r-functions and Q-function in Theorem 4.3, Eq(10)-(13). It is however non-trivial to theoretically extend our results to such setting where we need to address the issues of training instability on coupled iteration (note that this is an open problem even in the analysis of double Q-learning) as well as the related theory of SPE under function approximations. We have put this in our research agenda.
> >
> > > Revision: page 14, Section 7.

---

> > > ### Author Response · Authors · 2022-10-06
> > > **Response for Reviewer bU9Q (3)**
> > >
> > > ## References
> > > [Lin91c] Lin, L.J., 1991, July. Programming robots using reinforcement learning and teaching. In AAAI (pp. 781-786).
> > >
> > > [Lee19] Lee, S.Y., Sungik, C. and Chung, S.Y., 2019. Sample-efficient deep reinforcement learning via episodic backward update. Advances in Neural Information Processing Systems, 32.
> > >
> > > [Bai21] Bai, C., Wang, L., Han, L., Hao, J., Garg, A., Liu, P. and Wang, Z., 2021, July. Principled exploration via optimistic bootstrapping and backward induction. In International Conference on Machine Learning (pp. 577-587). PMLR.
> > >
> > > [LP21] Lesmana, N.S. and Pun, C.S., 2021. A Subgame Perfect Equilibrium Reinforcement Learning Approach to Time-inconsistent Problems. arXiv preprint arXiv:2110.14295.
> > >
> > > [Hasselt21] Hasselt, H., 2010. Double Q-learning. Advances in neural information processing systems, 23.
> > >
> > > [Xiong20] Xiong, H., Zhao, L., Liang, Y. and Zhang, W., 2020. Finite-time analysis for double Q-learning. Advances in neural information processing systems, 33, pp.16628-16638.
> > >
> > > [Zhao21] Zhao, L., Xiong, H. and Liang, Y., 2021. Faster Non-asymptotic Convergence for Double Q-learning. Advances in Neural Information Processing Systems, 34, pp.7242-7253.
> > >
> > > [Jin18] Jin, C., Allen-Zhu, Z., Bubeck, S. and Jordan, M.I., 2018. Is Q-learning provably efficient?. Advances in neural information processing systems, 31.
> > >
> > > [Jin20] Jin, C., Yang, Z., Wang, Z. and Jordan, M.I., 2020, July. Provably efficient reinforcement learning with linear function approximation. In Conference on Learning Theory (pp. 2137-2143). PMLR.

---

### Author Response · Authors · 2022-10-06
**Response to All Reviewers**

Dear reviewers,

Thank you for your detailed reviews and helpful comments. We have uploaded our revised manuscript and have colored and marked the parts we edited according to each reviewer as follows,
- Reviewer bU9Q: [R1], in *purple*
- Reviewer PRKT: [R2], in *blue*
- Reviewer agjL: [R3], in *orange*
- One or more reviewers: [R1-3] or [R2-3], in *teal*

We note correspondingly that the equation references used in our response below are to this revised version.

Thank you and regards,

Authors of Paper317

---

### Author Response · Authors · 2022-11-16
**Thank You and Camera-Ready Version**

Dear Reviewers and Action Editor,

We have uploaded the camera-ready version of our paper and made our GitHub repo public. Compared to the last revision, we have removed the color codes. Thank you again for your extensive reviews and for acknowledging our contributions. It was a great experience submitting to TMLR!

Best Regards,
Authors of Paper317

---

### Decision · Action_Editors · 2022-11-05

**Recommendation:** Accept as is

**Comment:**

All the reviewers found the problem studied in the paper interesting and the paper of good quality. They initially had questions and asked for some changes (mostly minor) and clarifications. The authors properly addressed all the questions and revised their paper according to the reviewers' suggestions. I vote for the acceptance and strongly recommend the authors to ensure that they address all the reviewers' comments in the final version of the paper.

**Audience:**

The findings of this paper are relevant to a subset of TMLR's audience.

**Claims And Evidence:**

The claims made in the submission are supported by convincing and clear evidence.